# Bridging the Distribution Gap to Harness Pre-trained Diffusion Priors for Super-Resolution

**Joonkyu Park[1] and Kyoung Mu Lee[1,2]**
[1]Dept. of ECE&ASRI, [2]IPAI, Seoul National University
`{jkpark0825,kyoungmu}@snu.ac.sr`

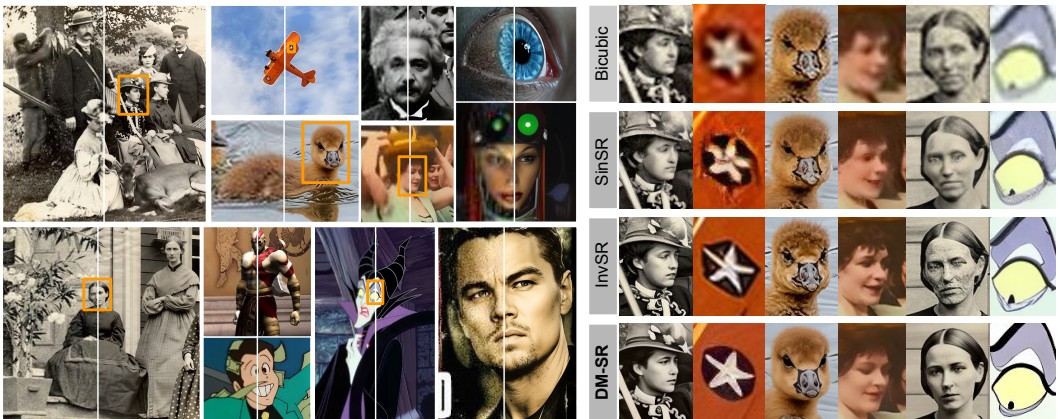

Figure 1: ×4 **super-resolution comparison on various images.** The left half of each image shows the bicubic upsampled input, and the right half shows the output from our DM-SR. Compared to previous methods, DM-SR produces the most perceptually pleasing results. (Zoom-in for best view)

## Abstract

Diffusion models, well recognized for their strong generative priors, have recently been increasingly applied to super-resolution (SR) tasks. However, as diffusion models are trained on Gaussian-corrupted natural images, the distribution gap between low-resolution (LR) inputs and the model's training distribution hinders direct inference. Prior works address this by conditioning on LR images, but their fine-tuning often weakens generative capability and requires multiple denoising steps. In this work, we present DM-SR, a novel framework that bridges this gap without modifying the pretrained diffusion model. We train an image encoder that maps LR inputs into the latent distribution aligned with the diffusion model's training space, preserving its full generative power. Furthermore, DM-SR adaptively predicts the appropriate noise level based on the degradation of each input, ensuring optimal alignment with the diffusion model's timestep-dependent distribution. Extensive experiments show that DM-SR achieves superior perceptual quality with a single-stage diffusion process, setting a new direction for efficient and high-fidelity SR with diffusion models.

## 1 Introduction

In the realm of low-level computer vision, single-image super-resolution (SISR or SR) is a fundamental task that aims to reconstruct high-resolution (HR) images from their low-resolution (LR) counterparts. Given the inherently ill-posed nature of this problem, with many plausible solutions, the practical goal of SR is to generate visually pleasing results from LR inputs Kim et al. (2016); Lim et al. (2017); Wang et al. (2018c); Ma et al. (2020); Ledig et al. (2017); Liang et al. (2022a); Umer & Micheloni (2020). Recently, inspired by the success of powerful diffusion models Ho et al. (2020); Rombach et al. (2022) in producing realistic images, several studies Li et al. (2022); Wu et al. (2024a); Dong et al. (2025); Yue et al. (2025); Wu et al. (2024b); Yang et al. (2024); Wang et al. (2024b) have explored leveraging diffusion models for SR tasks.

Specifically, diffusion models are trained to progressively denoise images, where the inputs are typically a combination of natural high-quality images and Gaussian noise. However, low-resolution images follow a substantially different distribution from this training setup, making it difficult to use them directly as inputs to the pretrained diffusion model. Consequently, prior diffusion-based SR methods Wang et al. (2024a); Lin et al. (2024); Yang et al. (2024); Wu et al. (2024b) still apply noise-corrupted HR images as direct inputs for denoising during training, using LR images as conditions Zhang et al. (2023a) to guide the diffusion process. On the other hand, during inference, where HR images are unavailable, they start the diffusion process from pure Gaussian noise. By leveraging the generative capabilities of pretrained diffusion models, these methods produce more realistic HR images compared to traditional non-diffusion-based SR approaches. However, the generative prior may be weakened during the fine-tuning process, and it often redundantly regenerates information already present in LR images, resulting in unrealistic outputs. This also leads to the necessity of multiple iterative diffusion steps, which increases overall computational cost.

To address these issues, later approaches propose directly using LR images directly as inputs to better preserve the information already present in them, while compressing the diffusion process into only a few steps (or even a single step). Among single-step diffusion-based SR methods, several works Wu et al. (2024a); Dong et al. (2025) distill the knowledge of diffusion models into SR networks, where the SR network is fine-tuned from a pretrained diffusion backbone. However, these methods rely on the diffusion prior only indirectly through the training objective Yin et al. (2024b;a), and fine-tuning the denoiser (*i.e.*, the SR network) compromises its original generative ability, preventing the SR network from fully retaining the pretrained generative prior. In contrast, *InvSR* Yue et al. (2025) avoids training the denoiser by predicting the noise under the assumption that noisy HR and LR images are indistinguishable. However, this assumption is *not strictly valid*: the indistinguishability depends on the input-preservation factor (*i.e.*, timestep). When the timestep is small, a large portion of the original signal is preserved, causing noisy HR and LR images to follow different distributions.

To address the above limitations, we propose Distribution Matching Super-Resolution (DM-SR), a simple yet effective framework. DM-SR is built on a fundamental yet important idea: if pretrained diffusion models are optimized to denoise samples drawn from a noise-corrupted image distribution, why not directly transform LR images into that same distribution? Based on this, we focus on aligning LR images with the distribution that pretrained diffusion models are familiar with. Specifically, we train only an image encoder to map LR images into this diffusion-compatible distribution, which consists of both the image and noise components, where the image and noise parts are differentiated through pretrained diffusion models. Moreover, as the diffusion model's familiar distribution varies across different noise levels (*i.e.*, timesteps), we adaptively map highly degraded LR images to the distributions with higher noise levels accordingly and vice versa. This reflects the fact that the generative power of the diffusion model is more crucial for highly degraded samples.

By fully leveraging the generative prior of diffusion models, our DM-SR can produce highly photo-realistic results on a wide range of benchmark datasets, including both synthetically degraded Deng et al. (2009) and real degraded Wei et al. (2020); Cai et al. (2019); Yue et al. (2023) LR images. Additionally, extensive ablation studies further validate the effectiveness and generalizability of our DM-SR. Our contributions can be summarized as follows:

- We propose DM-SR, which leverages a pretrained diffusion model for super-resolution without modifying the diffusion model. This bypasses the need for fine-tuning the diffusion model and enables full utilization of its generative prior.

- We introduce an encoder that maps LR inputs into the diffusion model's native distribution, dynamically aligned to the timestep-dependent prior. This ensures that the generative power of the diffusion model is more leveraged for highly degraded samples.

- DM-SR achieves state-of-the-art perceptual quality with a single diffusion step, offering both efficiency and fidelity. This is shown through extensive experiments, confirming the framework's effectiveness in achieving high perceptual quality.

## 2 RELATED WORKS

**Deep learning-based Super-Resolution.** Super-resolution (SR) constitutes a fundamental task in image restoration Park et al. (2022); Lee et al. (2024); Park et al. (2023a), as it aims to recover

fine-grained details from low-resolution images and plays a critical role in numerous real-world applications. After naïve SRCNN Dong et al. (2015), innovative CNN-based architectures such as VDSR Kim et al. (2016), EDSR Lim et al. (2017), and SRGAN Ledig et al. (2017) were introduced, opening a new horizon for deep-learning-based SR. With the rise of attention mechanisms Wang et al. (2018a); Vaswani et al. (2017), numerous architectural innovations have been proposed to further improve SR performance. RCAN Zhang et al. (2018b) introduced a channel attention mechanism, while SwinIR Liang et al. (2021a) and SwinFIR Zhang et al. (2022) leveraged Transformer Vaswani et al. (2017)-based architectures to enhance super-resolution capabilities. However, most of these methods are trained on datasets generated through simple downsampling methods (*i.e.*, bicubic or bilinear interpolation) Timofte et al. (2017); Wang et al. (2018b); Lim et al. (2017), making them less effective when dealing with real-world degraded images. To overcome this limitation, later approaches incorporated degradation-based data augmentation techniques (*e.g.*, adding noise, JPEG compression, defocus blur, and varying resizing) Wang et al. (2021); Zhang et al. (2021a); Yue et al. (2023) to better simulate the complex degradations encountered in real scenarios.

**GAN-based Super-Resolution.** Earlier SR methods trained their networks by minimizing pixel-wise losses between the super-resolved and high-resolution images. However, such losses often fail to capture perceptual quality, leading to visually unsatisfying results. To address this, subsequent works Wang et al. (2018c); Umer & Micheloni (2020); Pan et al. (2021); Wei et al. (2021); Castillo et al. (2021); Ma et al. (2020); Zhang et al. (2021b); Liang et al. (2022a); Park et al. (2023b) introduced perceptual losses, such as VGG loss and adversarial Goodfellow et al. (2014) loss. Starting with ESRGAN Wang et al. (2018c), which first applied a GAN framework to SR by jointly training a generator (*i.e.*, SR network) and a discriminator, numerous later approaches have explored improved GAN-based strategies. Specifically, DASR Liang et al. (2022b) employed wavelet transforms to focus the GAN on high-frequency components, while LDL Liang et al. (2022a) introduced a discriminator that distinguishes between visual artifacts and realistic details to better regularize adversarial training. SFT-GAN Wang et al. (2018b) incorporated explicit semantic information (*e.g.*, segmentation maps) into the GAN framework, and CAL-GAN Park et al. (2023b) used local discriminators tailored to content regions through a mixture of classifiers Jacobs et al. (1991). While these methods achieve more visually realistic SR results compared to those relying solely on pixel-wise losses, they still struggle to generate natural and coherent details.

**Diffusion-based Super-Resolution.** Driven by the powerful generative priors of diffusion models (DDPMs) Ho et al. (2020), pretrained diffusion models have been widely adopted across various tasks, including SR task. Early methods Kawar et al. (2022); Chung et al. (2022); Wang et al. (2023b) correlated intermediate diffusion steps with the LR input through variational distribution modeling or null-space decomposition. While effective, they require solving additional optimization problems at every diffusion step, resulting in significant computational overhead. To address this, later works Wang et al. (2024a); Yu et al. (2024); Wu et al. (2024b); Yang et al. (2024); Wang et al. (2024b); Lin et al. (2024) have explored fine-tuning pretrained diffusion models for SR tasks. Instead of updating the full model, which is computationally intensive and may harm the generative prior, most adopt the ControlNet Zhang et al. (2023a) architecture, fine-tuning only a few additional layers. Specifically, StableSR Wang et al. (2024a) fine-tunes a time-aware encoder, SeeSR Wu et al. (2024b) learns degradation-aware prompts, and Yang et al. (2024); Yu et al. (2024) train degradation-aware encoders to provide robust conditions.

Despite their improvements, aforementioned approaches still rely on a multi-step diffusion process starting from pure noise, which leads to several drawbacks: high computational cost, redundant regeneration of information already present in the LR image, and sensitivity to the choice of initial noise. To mitigate this, more recent methods Dong et al. (2025); Wu et al. (2024a); Yue et al. (2025) have proposed single-step SR frameworks that either train an SR network (student) to mimic the output distribution of a pretrained diffusion model (teacher) or perform noise prediction at fixed timesteps under the assumption that noisy LR and HR images are identical. While these approaches reduce randomness and computational cost, the student networks fine-tuned from pretrained diffusion models often degrade the original generative prior, leading to lower perceptual quality than full diffusion models. Similarly, applying the same noise ratio (*i.e.*, timestep) to all LR samples under this incorrect assumption can further reduce perceptual quality.

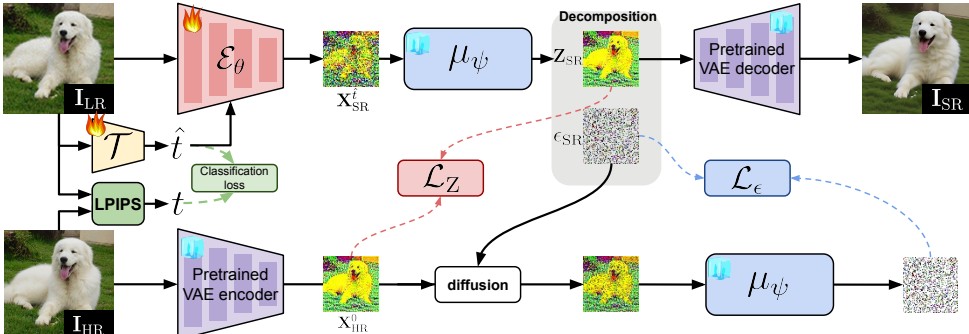

Figure 2: **The overall architecture of Distribution-Matching SR (DM-SR),** While keeping the pretrained denoiser $\mu_\psi$ fixed, we train the image encoder $\mathcal{E}_\theta$ to estimate $\mathbf{X}_{\text{SR}}^{\hat{t}}$ from the LR image $\mathbf{I}_{\text{LR}}$ and the predicted timestep $\hat{t}$. DM-SR then leverages $\mu_\psi$ to produce a realistic latent $\mathbf{Z}_{\text{SR}}$, which is fed into the pretrained VAE decoder to generate the final SR image $\mathbf{I}_{\text{SR}}$. During training, the encoder $\mathcal{E}_\theta$ is optimized using both image ($\mathcal{L}_{\text{Z}}$) and noise ($\mathcal{L}_\epsilon$) losses.

## 3 PROPOSED METHOD

Figure 2 shows the overall pipeline of our DM-SR. Unlike previous diffusion-based SR approaches that either train control modules Wang et al. (2024a); Yu et al. (2024); Wu et al. (2024b); Yang et al. (2024); Lin et al. (2024) or fine-tune an entire SR network Dong et al. (2025); Wu et al. (2024a), DM-SR trains only an image encoder $\mathcal{E}_\theta$. The encoder is trained to map LR inputs $\mathbf{I}_{\text{LR}}$ into the distribution that the pretrained diffusion model is familiar with, enabling the generation of super-resolved outputs $\mathbf{I}_{\text{SR}}$ without fine-tuning the pretrained diffusion model's denoiser $\mu_\psi$.

### 3.1 TIMESTEP ESTIMATION

Since pretrained diffusion models $\mu_\psi$ are trained to recover the original images from the corresponding Gaussian-corrupted ones, they are inherently designed to handle mixtures of realistic images and Gaussian noise. However, the proportion of image and noise components that the pretrained diffusion model is familiar with varies depending on the timestep:

$$\mathbf{I}_{\text{HR}}^t = \sqrt{\bar{\alpha}_t}\mathbf{I}_{\text{HR}}^0 + \sqrt{1 - \bar{\alpha}_t}\epsilon, \tag{1}$$

where $\mathbf{I}_{\text{HR}}^t$ denotes the noisy HR image at timestep $t$, derived from the clean HR image $\mathbf{I}_{\text{HR}}^0$ with added random Gaussian noise $\epsilon \sim \mathcal{N}(\mathbf{0}, \mathbf{I})$. The coefficient $\bar{\alpha} = \{\bar{\alpha}_1, \bar{\alpha}_2, \cdots, \bar{\alpha}_N\}$ represents the cumulative product of the noise schedule, defined as $\bar{\alpha}_t = \prod_{s=1}^{t}(1 - \beta_s)$, where $\beta_*$ is determined by a predefined variance scheduler.

Therefore, instead of arbitrarily encoding the LR image $\mathbf{I}_{\text{LR}}$ into a distribution corresponding to any diffusion timestep, we first predict an appropriate timestep $\hat{t}$ that aligns with the degradation level of the input. As smaller timesteps in the diffusion model preserve more input contents, we assign smaller timesteps to less degraded images and vice versa. To do this, as shown in Figure 2, we use the normalized LPIPS Zhang et al. (2018a) score $\in [0, 500]$ between $\mathbf{I}_{\text{LR}}$ and the corresponding HR image $\mathbf{I}_{\text{HR}}$ as the ground truth timestep $t$. The timestep estimator $\mathcal{T}$ is then trained to predict $\hat{t}$ to match this ground truth. For details of the timestep estimator and the validation of using LPIPS scores as the ground-truth timestep, please refer to the Appendix A.1 and Section 4.5, respectively.

### 3.2 ENCODING LR IMAGES

Given an LR image $\mathbf{I}_{\text{LR}}$ and its predicted timestep $\hat{t}$, image encoder $\mathcal{E}_\theta$ maps them into a distribution aligned with what the pretrained diffusion model is familiar with. Inspired by stable diffusion models Rombach et al. (2022), which significantly reduce computational costs, we operate encoding process into the latent space. Specifically, we encode $\mathbf{I}_{\text{LR}}$ into a latent representation $\mathbf{X}_{\text{SR}}^{\hat{t}}$, as shown in Figure 2, aiming to match the distribution of the corresponding noisy HR latent at timestep $\hat{t}$, $\mathbf{X}_{\text{HR}}^{\hat{t}}$.

Here, the noisy HR latent $\mathbf{X}_{\text{HR}}^{\hat{t}}$ can be obtained by applying Equation 1 in the latent space, where the image domain $\mathbf{I}_{\text{HR}}^{*}$ and timestep $t$ are replaced with the latent representation $\mathbf{X}_{\text{HR}}^{*}$ and the predicted timestep $\hat{t}$, respectively. Here, $\mathbf{X}_{\text{HR}}^{0}$ is obtained by encoding $\mathbf{I}_{\text{HR}}$ using the pretrained VAE Kingma & Welling (2013). For the architecture of $\mathcal{E}_{\theta}$, we adopt the pretrained VAE encoder and inject features derived from the predicted timestep $\hat{t}$ into each intermediate layer via single linear layers, following the ControlNet Zhang et al. (2023a) design. For details of the encoder architecture, please refer to the Appendix A.1.

## 3.3 Decomposing Latent Representation $\mathbf{X}_{\text{SR}}^{\hat{t}}$

While the image encoder $\mathcal{E}_{\theta}$ maps the LR image $\mathbf{I}_{\text{LR}}$ into a latent representation $\mathbf{X}_{\text{SR}}^{\hat{t}}$, our goal is for this representation to align with the distribution of $\mathbf{X}_{\text{HR}}^{\hat{t}}$—a mixture of latent features from the HR image and Gaussian noise. However, since $\mathbf{X}_{\text{HR}}^{\hat{t}}$ depends on a randomly sampled noise $\epsilon$, as in Equation 1, directly supervising $\mathbf{X}_{\text{SR}}^{\hat{t}}$ to match it is non-trivial. To address this, we decompose $\mathbf{X}_{\text{SR}}^{\hat{t}}$ into two components: the image part $\mathbf{Z}_{\text{SR}}$ and the noise part $\epsilon_{\text{SR}}$, allowing for more targeted supervision. This decomposition is performed using the pretrained diffusion denoiser $\mu_{\psi}$ from the stable diffusion model, as follows:

$$
\begin{aligned}
\epsilon_{\text{SR}} &= \mu_{\psi}\left(\mathbf{X}_{\text{SR}}^{\hat{t}}, \hat{t}\right), \\
\mathbf{Z}_{\text{SR}} &= \frac{1}{\sqrt{\bar{\alpha}_{\hat{t}}}}\left(\mathbf{X}_{\text{SR}}^{\hat{t}} - \sqrt{1 - \bar{\alpha}_{\hat{t}}}\,\epsilon_{\text{SR}}\right),
\end{aligned}
\tag{2}
$$

with the assumption that the pretrained diffusion denoiser $\mu_{\psi}$ estimates the noise component, and the remaining part corresponds to the latent representation of a realistic image. Here, we use fixed text prompts [1] as the text condition for the denoiser $\mu_{\psi}$. We then supervise the decomposed components, $\mathbf{Z}_{\text{SR}}$ and $\epsilon_{\text{SR}}$, to ensure that their combination $\mathbf{X}_{\text{SR}}^{\hat{t}}$ aligns with the distribution of $\mathbf{X}_{\text{HR}}^{\hat{t}}$. Note that, during evaluation, we decode $\mathbf{Z}_{\text{SR}}$ to super-resolved image $\mathbf{I}_{\text{SR}}$ using the pretrained VAE decoder.

## 3.4 Objectives for Image Part

For the image component $\mathbf{Z}_{\text{SR}}$, we supervise it directly using the latent representation of the HR image $\mathbf{X}_{\text{HR}}^{0}$. Specifically, we apply a combination of L1 loss $\mathcal{L}_{\text{L1}}$, perceptual loss $\mathcal{L}_{\text{per}}$, adversarial loss $\mathcal{L}_{\text{adv}}$, and distribution matching loss $\mathcal{L}_{\text{dm}}$:

$$
\mathcal{L}_{\text{Z}} = \lambda_{\text{L1}}\mathcal{L}_{\text{L1}} + \lambda_{\text{per}}\mathcal{L}_{\text{per}} + \lambda_{\text{adv}}\mathcal{L}_{\text{adv}} + \lambda_{\text{dm}}\mathcal{L}_{\text{dm}},
\tag{3}
$$

where $\lambda_{\text{L1}} = 1.0$, $\lambda_{\text{per}} = 2.0$, $\lambda_{\text{adv}} = 0.1$, and $\lambda_{\text{dm}} = 0.5$ are empirically chosen hyperparameters of the corresponding loss functions. For the details about hyperparameters, please refer to the Appendix A.2. Here, for the perceptual loss, since the conventional VGG network Simonyan & Zisserman (2014) operates in the image domain, we follow Yue et al. (2025) and compute the loss in the latent space instead.

Regarding adversarial loss, we observe that image component $\mathbf{Z}_{\text{SR}}$ often appears realistic but exhibits significant content discrepancies from the ground truth HR images. To address this, unlike traditional discriminators Goodfellow et al. (2014) used in GAN-based SR methods, which naively distinguish between HR and SR images (or their latent representations), we design a discriminator $\mathcal{D}_{\phi}$ following the ControlNet Zhang et al. (2023a) architecture. Specifically, the discriminator $\mathcal{D}_{\phi}$ takes the LR image $\mathbf{I}_{\text{LR}}$ as a conditional input and distinguishes between either predicted image component $\mathbf{Z}_{\text{SR}}$ or the real latent $\mathbf{X}_{\text{HR}}^{0}$. For details of the discriminator architecture, please refer to the supplementary material. Our discriminator $\mathcal{D}_{\phi}$ is optimized by the following objective:

$$
\begin{aligned}
\min_{\mathcal{D}_{\phi}} &- \mathbb{E}_{\mathbf{X}_{\text{HR}}^{0} \sim p(\mathbf{X}_{\text{HR}}^{0})}[\log(\mathcal{D}_{\phi}(\mathbf{X}_{\text{HR}}^{0}, \mathbf{I}_{\text{LR}}))] \\
&- \mathbb{E}_{\mathbf{Z}_{\text{SR}} \sim p(\mathbf{Z}_{\text{SR}})}[\log(1 - \mathcal{D}_{\phi}(\mathbf{Z}_{\text{SR}}, \mathbf{I}_{\text{LR}}))].
\end{aligned}
\tag{4}
$$

Jointly, to fool the discriminator $\mathcal{D}_{\phi}$, the image encoder $\mathcal{E}_{\theta}$ is trained with the adversarial loss:

$$
\mathcal{L}_{\text{adv}} = -\mathbb{E}_{\mathbf{Z}_{\text{SR}} \sim p(\mathbf{Z}_{\text{SR}})}[\log(\mathcal{D}_{\phi}(\mathbf{Z}_{\text{SR}}, \mathbf{I}_{\text{LR}}))],
\tag{5}
$$

---

[1] "High-quality, high-contrast, photo-realistic, clean, sharp, highly-detailed, perfect, 8k, ultra HD"

where gradients are propagated only to the encoder $\mathcal{E}_\theta$; both the diffusion denoiser $\mu_\psi$ and the discriminator $\mathcal{D}_\phi$ are kept fixed during this update. This enables the discriminator to assess not only the realism of the samples but also their alignment with the input LR images $\mathbf{I}_{\text{LR}}$.

Furthermore, motivated by distillation techniques Yin et al. (2024b;a) in diffusion models, we introduce a distribution matching loss $\mathcal{L}_{\text{dm}}$ to align the generated image part latent $\mathbf{Z}_{\text{SR}}$ with the ideal target $\mathbf{X}_{\text{HR}}^0$. To achieve this, we apply the same noise scheduler to both $\mathbf{Z}_{\text{SR}}$ and $\mathbf{X}_{\text{HR}}^0$ and encourage the pretrained denoiser $\mu_\psi$ to predict consistent score functions from their noisy versions. Specifically, following Equation 1, we first corrupt both latents, $\mathbf{Z}_{\text{SR}}$ and $\mathbf{X}_{\text{HR}}^0$, using shared random Gaussian noise $\epsilon$ to obtain:

$$\tilde{\mathbf{Z}}_{\text{SR}}^{\hat{t}} = \sqrt{\bar{\alpha}_t}\mathbf{Z}_{\text{SR}} + \sqrt{1 - \bar{\alpha}_t}\epsilon, \quad \tilde{\mathbf{X}}_{\text{HR}}^{\hat{t}} = \sqrt{\bar{\alpha}_t}\mathbf{X}_{\text{HR}}^0 + \sqrt{1 - \bar{\alpha}_t}\epsilon, \tag{6}$$

We then define the distribution matching loss $\mathcal{L}_{\text{dm}}$ as:

$$\mathcal{L}_{\text{dm}} = \mathbb{E}_{\mathbf{Z}_{\text{SR}}, \tilde{\mathbf{X}}_{\text{HR}}^{\hat{t}}, \hat{t}}\left[\left(\mu_\psi(\tilde{\mathbf{Z}}_{\text{SR}}^{\hat{t}}, \hat{t}) - \mu_\psi(\tilde{\mathbf{X}}_{\text{HR}}^{\hat{t}}, \hat{t})\right)\frac{d\mathcal{E}}{d\theta}\right], \tag{7}$$

where the expectation is computed across $\hat{t} \in [1, 500]$. Note that only the encoder $\mathcal{E}_\theta$ is updated while the denoiser $\mu_\psi$ remains fixed.

### 3.5 Objectives for Noise Part

While the image part loss $\mathcal{L}_Z$ guides the encoder to produce image part $\mathbf{Z}_{\text{SR}}$ close to $\mathbf{X}_{\text{HR}}^0$ and noise that is easier for the pretrained denoiser $\mu_\psi$ to remove, further constraints are needed to regularize the noise itself directly. Considering that diffusion models progressively transform an image into noise, there exists an optimal noise for each image such that, when added to $\mathbf{X}_{\text{HR}}^0$ and passed through $\mu_\psi$, it reconstructs the original HR image Qi et al. (2024); Yue et al. (2025). Based on this, we corrupt the HR latent $\mathbf{X}_{\text{HR}}^0$ using the predicted noise $\epsilon_{\text{SR}}$, and encourage $\epsilon_{\text{SR}}$ to approximate the optimal noise that enables reconstruction of the original HR latent. We define the noise loss $\mathcal{L}_\epsilon$ as:

$$\mathcal{L}_\epsilon = \mathbb{E}_{\mathbf{X}_{\text{HR}}^0, \epsilon_{\text{SR}}, \hat{t}}\left[\left(\mu_\psi(\sqrt{\bar{\alpha}_t}\mathbf{X}_{\text{HR}}^0 + \sqrt{1 - \bar{\alpha}_t}\epsilon_{\text{SR}}, \hat{t}) - \epsilon_{\text{SR}}\right)\frac{d\mathcal{E}}{d\theta}\right]. \tag{8}$$

Finally, we combine the described image and noise part objectives to define the total loss $\mathcal{L}_{\text{tot}}$ as:

$$\mathcal{L}_{\text{tot}} = \mathcal{L}_Z + \mathcal{L}_\epsilon. \tag{9}$$

## 4 Experiments

### 4.1 Datasets and Metrics

**Datasets.** For training DM-SR, we adopt DF2K Timofte et al. (2017) and LSDIR Li et al. (2023). For evaluation, we test DM-SR on various benchmarks, including synthetic LR images from ImageNet Deng et al. (2009) and real-world LR images from DRealSR Wei et al. (2020), RealSR Cai et al. (2019), and RealSet80 Yue et al. (2023). Here, for ImageNet, we follow the degradation protocol of InvSR Yue et al. (2025).

**Metrics.** To evaluate the photo-realistic quality of our super-resolved images, we primarily utilize non-reference-based perceptual metrics such as BRISQUE Mittal et al. (2012), LIQE Zhang et al. (2023b), CLIP-IQA Wang et al. (2023a), TOPIQ (NR) Chen et al. (2023), NIMA Talebi & Milanfar (2018), MANIQA Yang et al. (2022), and MUSIQ Liang et al. (2021b). Furthermore, we also include reference-based metrics like PSNR, SSIM, LPIPS Zhang et al. (2018a), DISTS Ding et al. (2020), and TOPIQ (FR) Chen et al. (2023) for completeness. Here, all metrics are evaluated in the standard RGB space, except for SSIM, which is computed on the luminance (Y) channel.

**Experimental setups.** Like previous diffusion-based SR approaches Wu et al. (2024a); Dong et al. (2025), we initialize our encoder $\mathcal{E}_\theta$ with the pretrained VAE encoder. For the diffusion denoiser $\mu_\psi$, we adopt the pretrained SD-Turbo Sauer et al. (2024) due to its efficiency in generating realistic outputs in a single step. We fine-tune only the encoder $\mathcal{E}_\theta$ using the loss in Equation 9. During training, we use LR and HR patch pairs ($\in \mathbb{R}^{512 \times 512 \times 3}$) with a batch size of 16, where LR patches are bicubic upsampled from $4\times$ lower resolution, following the ResShift pipeline Yue et al. (2023). The model is trained using the AdamW optimizer Loshchilov & Hutter (2017) for 300k iterations, starting with a learning rate of $1 \times 10^{-4}$, which is reduced by half every 100k steps.

Table 1: ×4 SR non-reference metrics comparison on various benchmark datasets. **Best** numbers are denoted with **bold**.

| | Method | Dataset | BRISQUE↓ | LIQE↑ | CLIPIQA↑ | TOPIQ (NR)↑ | NIMA↑ | MANIQA↑ | MUSIQ↑ |
|---|---|---|---|---|---|---|---|---|---|
| **Synthetic** | RealESRGAN Wang et al. (2021) | ImageNet | 22.888 | 3.841 | 0.509 | 0.526 | 4.695 | 0.360 | 64.821 |
| | StableSR-50 Wang et al. (2024a) | | 19.067 | 4.094 | 0.631 | 0.594 | 4.900 | 0.418 | 68.247 |
| | SinSR-1 Wang et al. (2024b) | | 15.746 | 3.940 | 0.661 | 0.613 | 4.893 | 0.444 | 67.691 |
| | OSEDiff-1 Wu et al. (2024a) | | **12.032** | 4.561 | 0.678 | 0.630 | 5.097 | 0.459 | 71.750 |
| | *InvSR*-1 Yue et al. (2025) | | 15.139 | 4.560 | 0.711 | 0.630 | 5.225 | 0.469 | 72.382 |
| | **DM-SR-1 (Ours)** | | 13.427 | **4.699** | **0.785** | **0.712** | **5.492** | **0.633** | **73.856** |
| **Real** | RealESRGAN | DRealSR | 34.767 | 2.926 | 0.451 | 0.462 | 4.326 | 0.344 | 54.277 |
| | StableSR-50 | | 34.060 | 2.520 | 0.497 | 0.432 | 4.221 | 0.324 | 51.242 |
| | SinSR-1 | | 19.012 | 3.126 | 0.650 | 0.520 | 4.505 | 0.392 | 55.677 |
| | OSEDiff-1 | | 23.477 | 3.940 | 0.696 | 0.600 | 4.676 | 0.466 | 64.687 |
| | *InvSR*-1 | | **18.660** | 4.056 | 0.713 | 0.611 | 4.857 | 0.467 | 65.998 |
| | **DM-SR-1 (Ours)** | | 21.281 | **4.487** | **0.753** | **0.691** | **5.216** | **0.588** | **69.038** |
| | RealESRGAN | RealSR | 31.768 | 3.357 | 0.448 | 0.515 | 4.655 | 0.374 | 60.374 |
| | StableSR-50 | | 30.874 | 3.153 | 0.525 | 0.515 | 4.684 | 0.387 | 61.876 |
| | SinSR-1 | | 21.646 | 3.127 | 0.617 | 0.518 | 4.642 | 0.403 | 60.391 |
| | OSEDiff-1 | | 20.501 | 4.068 | 0.669 | 0.625 | 4.894 | 0.472 | 69.090 |
| | *InvSR*-1 | | 20.931 | 4.038 | 0.678 | 0.590 | 5.094 | 0.455 | 68.537 |
| | **DM-SR-1 (Ours)** | | **19.953** | **4.485** | **0.732** | **0.702** | **5.374** | **0.601** | **71.489** |
| | RealESRGAN | Realset 80 | 18.293 | 3.797 | 0.598 | 0.533 | 5.219 | 0.388 | 64.638 |
| | StableSR-50 | | 28.164 | 2.976 | 0.558 | 0.481 | 5.041 | 0.388 | 59.503 |
| | SinSR-1 | | 11.196 | 3.586 | 0.728 | 0.570 | 5.200 | 0.443 | 64.000 |
| | OSEDiff-1 | | 19.423 | 4.130 | 0.703 | 0.607 | 5.276 | 0.473 | 69.204 |
| | *InvSR*-1 | | **11.343** | 4.291 | 0.727 | 0.623 | 5.459 | 0.466 | 69.798 |
| | **DM-SR-1 (Ours)** | | 11.714 | **4.652** | **0.797** | **0.707** | **5.575** | **0.600** | **70.616** |

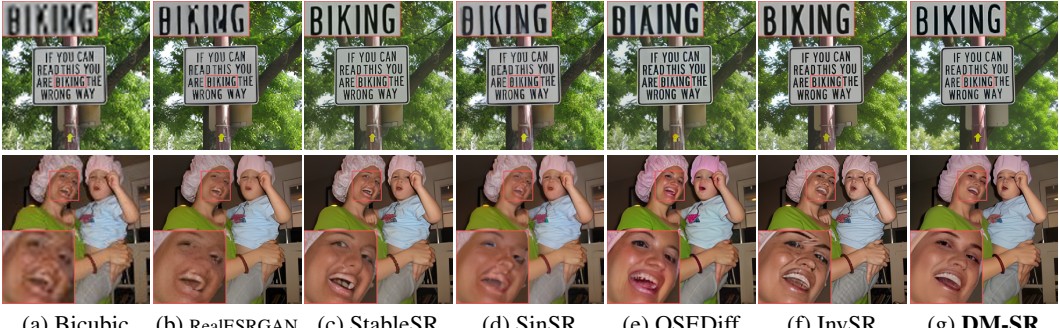

(a) Bicubic    (b) RealESRGAN    (c) StableSR    (d) SinSR    (e) OSEDiff    (f) InvSR    (g) **DM-SR**

Figure 3: ×4 **super-resolution qualitative comparison on various benchmark dataset. (Zoom-in for best view)** Please refer to our supplementary material for more extensive visual comparisons.

## 4.2    COMPARISON WITH STATE-OF-THE-ART METHODS

Table 1 compares DM-SR against prior SR methods, including RealESRGAN Wang et al. (2021) (a non-diffusion model), StableSR Wang et al. (2024a) (an iterative method using 50 DDPM Ho et al. (2020) steps), and recent single-step diffusion-based SR approaches Wang et al. (2024b); Wu et al. (2024a); Yue et al. (2025). As shown, our DM-SR consistently outperforms others in perceptual quality on both synthetic and real LR images. Moreover, Table 2 presents reference-based metrics. While DM-SR may not always achieve the best scores in reference metrics, the overall results in Tables 1 and 2 show that DM-SR maintains competitive distortion performance while delivering superior perceptual quality, highlighting DM-SR's favorable balance in the perceptual-distortion trade-off. Specifically, while DM-SR attains reference-based performance comparable to OSEDiff Wu et al. (2024a), it delivers notably superior perceptual quality.

Moreover, Table 2 compares the efficiency of DM-SR with previous super-resolution methods in terms of the number of trainable parameters and runtime. As shown, the GAN-based method Wang et al. (2021) achieves the best efficiency due to its lightweight design, but fails to deliver perceptually pleasing results, as shown in Table 1. Among diffusion-based methods, DM-SR maintains a comparable number of trainable parameters and achieves the fastest runtime, while still delivering outstanding perceptual performance, as shown in Table 1.

Figure 3 further highlights the perceptual superiority of our DM-SR, showing that our method produces sharper and more accurate reconstructions across diverse benchmarks. Specifically, single-step diffusion models Wang et al. (2024b); Wu et al. (2024a); Yue et al. (2025) fail to recover

Table 2: **(Left) × 4 SR reference metrics comparison on RealSR dataset. (Right) Efficiency Comparison of DM-SR with previous SR methods on SR task.** Specifically, we upsample images of size $\mathbb{R}^{128 \times 128 \times 3}$ using a single NVIDIA A100 GPU to measure the runtime of each method.

| Method | PSNR$_\uparrow$ | SSIM$_\uparrow$ | LPIPS$_\downarrow$ | DISTS$_\downarrow$ | TOPIQ (FR)$_\uparrow$ | # Params (M)$_\downarrow$ | Runtime (ms)$_\downarrow$ |
|---|---|---|---|---|---|---|---|
| RealESRGAN | 26.312 | 0.777 | 0.282 | 0.209 | 0.517 | 16.70 | 65 |
| StableSR-50 | 27.064 | 0.788 | 0.278 | 0.212 | 0.511 | 152.70 | 3460 |
| SinSR-1 | 25.851 | 0.717 | 0.368 | 0.248 | 0.476 | 118.58 | 140 |
| OSEDiff-1 | 25.938 | 0.755 | 0.297 | 0.216 | 0.527 | 8.50 | 177 |
| *InvSR*-1 | 23.740 | 0.680 | 0.354 | 0.246 | 0.431 | 33.84 | 117 |
| **DM-SR-1 (Ours)** | 24.984 | 0.710 | 0.317 | 0.218 | 0.470 | 34.16 | 92 |

Table 3: **Comparison of DM-SR on Realset80 with various timesteps.** Our final model adaptively predicts $\hat{t} \in [0, 500]$ from $\mathbf{I}_{\text{LR}}$ instead of relying on fixed timesteps.

| timestep | LIQE$_\uparrow$ | CLIPIQA$_\uparrow$ | TOPIQ (NR)$_\uparrow$ | NIMA$_\uparrow$ | MUSIQ$_\uparrow$ |
|---|---|---|---|---|---|
| 500 | 4.023 | 0.706 | 0.610 | 5.246 | 69.078 |
| 300 | 4.433 | 0.731 | 0.634 | 5.444 | 69.802 |
| 250 | 4.471 | 0.733 | 0.668 | 5.407 | 69.990 |
| 150 | 4.314 | 0.725 | 0.633 | 5.370 | 69.706 |
| 50 | 3.994 | 0.681 | 0.600 | 5.201 | 68.982 |
| $\hat{t}$ **(Ours)** | **4.652** | **0.797** | **0.707** | **5.575** | **70.616** |

Table 4: **Comparison of DM-SR on Realset80 with various number of steps.** Despite being controllable, single-step application still produces high-quality results.

| # of steps | LIQE$_\uparrow$ | CLIPIQA$_\uparrow$ | TOPIQ (NR)$_\uparrow$ | NIMA$_\uparrow$ | MUSIQ$_\uparrow$ |
|---|---|---|---|---|---|
| 20 | 4.152 | 0.729 | 0.737 | 5.081 | 67.800 |
| 10 | 4.503 | 0.762 | 0.743 | 5.531 | 68.135 |
| 5 | 4.773 | 0.768 | **0.749** | **5.756** | 69.948 |
| 2 | **4.799** | 0.778 | 0.744 | 5.692 | 69.435 |
| **1 (Ours)** | 4.652 | **0.797** | 0.707 | 5.575 | **70.616** |

Table 5: **Comparison of DM-SR on Realset80 with various ground truth for the timesteps.** Our final model utilize normalized LPIPS score $\in [0, 500]$ for the ground turth for the timesteps.

| ground truth | LIQE$_\uparrow$ | CLIPIQA$_\uparrow$ | TOPIQ (NR)$_\uparrow$ | NIMA$_\uparrow$ | MUSIQ$_\uparrow$ |
|---|---|---|---|---|---|
| $min^*$ | 3.848 | 0.596 | 0.563 | 5.248 | 65.630 |
| Pixel distance (L1) | 4.470 | 0.782 | 0.694 | 5.463 | 70.328 |
| SSIM | 4.628 | **0.799** | 0.703 | 5.572 | 70.352 |
| **LPIPS (Ours)** | **4.652** | 0.797 | **0.707** | **5.575** | **70.616** |

the letter (*e.g.*, "K" in "BIKING" in the first row of Figure 3) whereas our DM-SR successfully reconstructs them, providing better readability. Additionally, prior methods tend to generate severe artifacts on the face regions (*e.g.*, in the second row of Figure 3), while our DM-SR produces fewer artifacts and delivers significantly more perceptually pleasing results.

## 4.3 Effect of Adaptive Timestep $\hat{t}$

Table 3 compares the performance of DM-SR under different timestep settings. While ours adaptively predicts the timestep $\hat{t}$ based on the degradation level of the input image, the variants use fixed timesteps regardless of input quality. As shown, our adaptive approach consistently achieves the best perceptual SR performance, validating the effectiveness of our timestep estimator $\mathcal{T}$. Interestingly, among the fixed-timestep settings, larger timesteps generally lead to better results than smaller ones. This may be because smaller timesteps tend to preserve more low-resolution content, which often fails to align with the true high-resolution distribution. In contrast, larger timesteps allow the generative model more freedom to intervene during super-resolution, resulting in higher perceptual quality.

## 4.4 Discussion on Number of Steps

Table 4 shows a comparison of DM-SR performance using different numbers of sampling steps. For multiple steps, we adopt the DDIM Song et al. (2020) sampling, which deterministically generates the SR images without stochastic noise injection. During multi-step sampling, the same estimated noise $\epsilon_{\text{SR}}$ is reused to form the input for each subsequent iteration. In contrast, the timesteps for later iterations are linearly scheduled in proportion to the initially predicted timestep. Among the tested configurations, the perceptual quality varies across 1, 2, and 5-step settings, indicating that a small number of iterative refinements can still be effective. However, using 10 sampling steps consistently results in the worst perceptual scores across all evaluation metrics. This is likely due to the use of SD-Turbo as the underlying denoiser in our DM-SR framework. While SD-Turbo is capable of strong few-step denoising (*e.g.*, 1-5 steps), applying it over an excessive number of iterations can lead to over-smoothing, ultimately harming perceptual quality.

Table 6: **Comparison of DM-SR with various objective combinations.**

Table 7: **Comparison of DM-SR using fixed versus extracted text prompts.** Here, we use LLaVA to extract text prompts from low-quality inputs.

| $\mathcal{L}_{adv}$ | $\mathcal{L}_{dm}$ | $\mathcal{L}_{\epsilon}$ | LIQE↑ | CLIPIQA↑ | TOPIQ (NR)↑ | NIMA↑ | MUSIQ↑ |
|---|---|---|---|---|---|---|---|
| ✗ | ✗ | ✗ | 3.643 | 0.726 | 0.575 | 5.191 | 64.153 |
| ✓ | ✗ | ✗ | 4.579 | 0.779 | 0.694 | 5.480 | 69.947 |
| ✗ | ✓ | ✗ | 4.171 | 0.756 | 0.614 | 5.308 | 69.089 |
| ✗ | ✗ | ✓ | 4.195 | 0.756 | 0.615 | 5.397 | 69.189 |
| ✓ | ✓ | ✓ | **4.652** | **0.797** | **0.707** | **5.575** | **70.616** |

| Prompt | LIQE↑ | CLIPIQA↑ | TOPIQ (NR)↑ | NIMA↑ | MUSIQ↑ |
|---|---|---|---|---|---|
| Fixed | 4.652 | 0.797 | 0.707 | **5.575** | 70.616 |
| Input-specific | **4.668** | **0.801** | **0.709** | 5.564 | **70.731** |

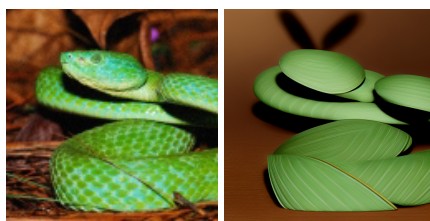
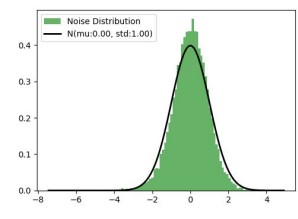
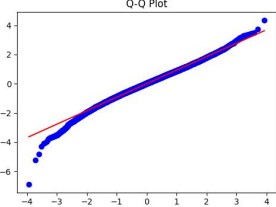

(a) $\mathbf{I}_{LR}$ and denoised results from $\epsilon_{SR}$         (b) Distribution of $\epsilon_{SR}$

Figure 4: **(Left) Validation of the generated noise $\epsilon_{SR}$.** Starting from the left LR inputs, encoder transforms them into $\mathbf{Z}_{SR}$ and $\epsilon_{SR}$. The diffusion process initialized with $\epsilon_{SR}$ produces outputs aligned with the input, as shown in the right images. **(Right) Distribution histogram and Q-Q plot of predicted noise $\epsilon_{SR}$.** Although it is not explicitly supervised to follow a Gaussian distribution, our predicted noise naturally approximates a Gaussian distribution.

### 4.5 GROUND TRUTH FOR THE PREDICTED TIMESTEPS

Without direct supervision for the predicted timestep $\hat{t}$, it is difficult to provide effective gradients to the timestep estimator $\mathcal{T}$, since the denoiser $\mu_{\psi}$ takes the timestep $\hat{t}$ as an integer input. While we use LPIPS as the ground truth (GT) for the predicted timestep, as described in Section 3.1, other alternatives are also possible. To validate the efficacy of using LPIPS as the GT, Table 5 shows comparisons with other choices. Specifically, we use normalized pixel-distance and SSIM score as alternative GT and observe that ours using LPIPS yields the best performance. Also, we attempt to supervise $\hat{t}$ by comparing SR results at $\hat{t} - 1$, $\hat{t}$, and $\hat{t} + 1$ and selecting the timestep that minimizes $||\mathbf{Z}_{SR} - \mathbf{X}_{HR}^{0}||$ ($min^{*}$ in Table 5). However, this approach produced unstable gradients and collapsed to a single timestep across inputs. Nonetheless, we acknowledge that there may exist more optimal formulations for defining the timestep GT, which we leave for future exploration.

### 4.6 EFFECT OF THE PROPOSED LOSS COMBINATION

Table 6 provides a quantitative analysis of the loss components: $\mathcal{L}_{adv}$, $\mathcal{L}_{dm}$, and $\mathcal{L}_{\epsilon}$. All variants are trained with the baseline losses $\mathcal{L}_{L1}$ and $\mathcal{L}_{per}$, with the three additional loss terms selectively included or omitted. Compared to the first row, which uses only $\mathcal{L}_{L1}$ and $\mathcal{L}_{per}$, adding the adversarial loss $\mathcal{L}_{adv}$ (second row) significantly improves perceptual performance. Although including either $\mathcal{L}_{dm}$ or $\mathcal{L}_{\epsilon}$ alone leads to only marginal gains compared to adding $\mathcal{L}_{adv}$, the full DM-SR model (last row), which incorporates all three additional loss terms, achieves the highest perceptual scores. This suggests that each loss provides complementary benefits beyond the baseline, leaving room for others to further enhance performance, thereby demonstrating strong synergy among the proposed loss terms.

### 4.7 EFFECT OF USED TEXT PROMPTS

While we use fixed text prompts in DM-SR framework, as described in Section 3.3, DM-SR does not update the denoiser $\mu_{\psi}$ during training. This allows DM-SR to be evaluated using different text prompts at test time without requiring any additional training. As the generative capability of the pretrained diffusion model vary depending on the text prompt, we investigate the impact of using different prompts. Table 7 shows a comparison between the two settings, where input-specific prompts are extracted from the LR images using LLaVA Li et al. (2024) [2]. As shown, adopting input-specific prompts consistently enhances perceptual quality, indicating that DM-SR has the potential to achieve even better results when provided with more informative or refined text prompts.

---

[2] lmms-lab/llava-onevision-qwen2-0.5b-ov

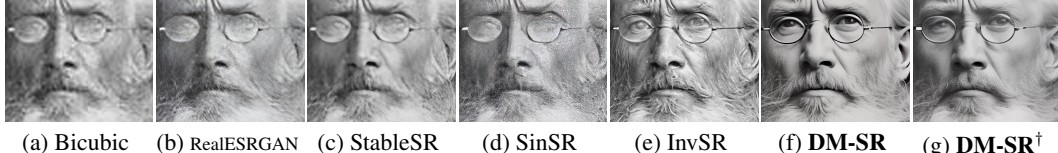

| (a) Bicubic | (b) RealESRGAN | (c) StableSR | (d) SinSR | (e) InvSR | (f) **DM-SR** | (g) **DM-SR**$^{\dagger}$ |

Figure 5: **Limitation of our DM-SR.** While prior methods successfully generate a gray pupil that closely matches the input image, our DM-SR produces a black pupil, prioritizing perceptual quality over strict distortion minimization. Here, while (f) DM-SR uses a fixed text prompt, (g) DM-SR$^{\dagger}$ employs a customized, input-specific prompt (*i.e.*, "Old man with a full white beard and mustache. He is wearing wire-rimmed glasses. His hair appears white, and his eyes are light-colored gray.").

### 4.8 DISCUSSION ON PREDICTED NOISE

In the DM-SR framework, unlike previous approaches that initialize with random noise, we transform the input image $\mathbf{I}_{\text{LR}}$ into two components: an image $\mathbf{Z}_{\text{SR}}$ and a noise $\epsilon_{\text{SR}}$. While the role of the image component is straightforward—since it directly contributes to forming the output image $\mathbf{I}_{\text{SR}}$—it is less intuitive to assess whether the predicted noise $\epsilon_{\text{SR}}$ reflects the content of the input image.

To validate that our encoder $\mathcal{E}_{\theta}$ adaptively generates a meaningful $\epsilon_{\text{SR}}$ that guides the subsequent denoiser $\mu_{\psi}$ to produce image details aligned with the input, we conduct a simple test, as shown in Figure 4a. Specifically, we initialize the diffusion process with $\epsilon_{\text{SR}}$ and apply 5 DDIM steps. As shown, the resulting image is well-aligned with the input $\mathbf{I}_{\text{LR}}$, indicating that $\epsilon_{\text{SR}}$ encodes relevant semantic details, and confirming that our encoder effectively conditions the noise for faithful SR.

For further validation of noise $\epsilon_{\text{SR}}$, we analyze whether the predicted noise follows a Gaussian distribution. As shown in Figure 4b, while we do not explicitly supervise $\epsilon_{\text{SR}}$ to match a Gaussian distribution, it aligns with a normal distribution, indicating that it exhibits a Gaussian-like structure.

## 5 LIMITATIONS

While our DM-SR achieves strong performance on perceptual metrics, as shown in Table 1, by effectively leveraging the generative priors of pretrained diffusion models, it does not attain the best distortion scores, as shown in Table 2. This trade-off may arise because the model prioritizes perceptual quality, which can occasionally lead to minor deviations from the exact appearance of the input LR image. For instance, in Figure 5, whereas other methods (Figures 5b-5e) generate a gray-colored pupil closely matching the LR input—resulting in lower distortion—DM-SR (Figure 5f) produces a black-colored pupil that aligns better with perceptual realism. This slight discrepancy indicates room for improvement, such as incorporating semantic-level color alignment or adopting adaptive, input-specific text prompts instead of a fixed one. Indeed, as shown in Figure 5g, simply using an input-specific text prompt that specifies the pupil color better preserves the original content. This observation suggests a promising direction for future work, extracting input-specific text prompts, to balance perceptual quality and distortion.

## 6 CONCLUSION

In this work, we propose DM-SR, a novel framework that effectively harnesses the generative power of pretrained diffusion models. By keeping the pretrained denoiser fixed and training only an image encoder to project LR inputs into a diffusion-aligned distribution, DM-SR fully preserves the pretrained generative prior without requiring model fine-tuning or iterative denoising steps. Moreover, our adaptive noise-level mapping adjusts the target distribution based on the degradation level of each image, improving perceptual metrics by a large margin across various degraded LR images. Extensive evaluations on synthetic and real-world benchmarks confirm that DM-SR achieves state-of-the-art perceptual quality. The framework also lays the groundwork for future extensions to other restoration tasks such as deblurring, denoising, and inpainting.

ACKNOWLEDGMENTS

This work was supported in part by the IITP grants [No. RS-2021-II211343, Artificial Intelligence Graduate School Program (Seoul National University), No. RS-2025-02303870, No.2022-0-00156] funded by the Korea government (MSIT).

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

## A APPENDIX

In this appendix, we provide discussions, details, and more visual results that could not be included in the main manuscript due to lack of space.

### A.1 ARCHITECTURES

**Timestep estimator $\mathcal{T}$.** We adopt a VGG-style architecture for the timestep estimator $\mathcal{T}$. Specifically, we utilize the first five convolutional blocks of VGG19 to extract features from the LR input $\mathbf{I}_{\text{LR}}$, which are then projected into a regressed feature vector $\mathbf{F}_{\hat{t}} \in \mathbb{R}^{500}$. To account for the continuous nature of timesteps, we do not apply a softmax classification over discrete bins. Instead, we compute the predicted timestep $\hat{t}$ as a weighted sum:

$$\hat{t} = sum(softmax(\mathbf{F}_{\hat{t}}) * [0, 1, \cdots, 488, 499]). \tag{10}$$

**Encoder $\mathcal{E}_\theta$.** While we use the pretrained VAE encoder as our encoder $\mathcal{E}_\theta$, we additionally incorporate the timestep $\hat{t}$ as an input, as depicted in Figure 2 of the main paper. Specifically, we project timestep $\hat{t}$ into four embedding vectors (in $\mathbb{R}^{128}$, $\mathbb{R}^{256}$, $\mathbb{R}^{512}$, and $\mathbb{R}^{512}$, respectively) using linear layers, and add these embeddings to the intermediate feature maps within the encoder. This allows our encoder to produce a timestep-aware latent representation $\mathbf{X}_{\text{SR}}^{\hat{t}}$.

**Discriminator $\mathcal{D}_\phi$.** As described in Section "Objectives for Image Part" of the main paper, our discriminator $\mathcal{D}_\phi$ is designed based on the ControlNet architecture. Here, to enable efficient training, we reduce the intermediate output channels of the UNet from [320, 640, 1280, 1280] to [64, 128, 256, 256]. The final output of the ControlNet-style discriminator is then passed through pooling and linear layers to produce a 1-dimensional output vector.

### A.2 HYPERPARAMETER FOR LOSS FUNCTIONS

As described in Section 3.4, we set the loss weights as follows: $\lambda_{\text{L1}} = 1.00$, $\lambda_{\text{per}} = 2.00$, $\lambda_{\text{adv}} = 0.10$, $\lambda_{\text{dm}} = 0.50$, and $\lambda_\epsilon = 1.00$. For the conventional loss terms, including $\mathcal{L}_{\text{L1}}$, $\mathcal{L}_{\text{per}}$, and $\mathcal{L}_{\text{adv}}$, we follow setting in *InvSR* Yue et al. (2025). For the newly introduced losses, $\mathcal{L}_{\text{dm}}$ and $\mathcal{L}_\epsilon$, we conduct ablation

Table 8: **Comparison of DM-SR on Realset80 with various hyperparameter configurations.** Our final model sets $\lambda_{\text{dm}} = 0.50$ and $\lambda_\epsilon = 1.00$.

| $\lambda_{\text{dm}}$ | 0.00 | 0.25 | 0.50 | 0.75 | 1.00 | 0.00 | 0.00 | 0.00 | 0.00 | 0.00 | 0.50 |
|---|---|---|---|---|---|---|---|---|---|---|---|
| $\lambda_\epsilon$ | 0.00 | 0.00 | 0.00 | 0.00 | 0.00 | 0.25 | 0.50 | 0.75 | 1.00 | 1.25 | 1.00 |
| MUSIQ$_\uparrow$ | 69.947 | 70.088 | **70.146** | 70.056 | 70.032 | 69.929 | 69.909 | 70.059 | **70.069** | 70.056 | 70.616 |

studies by varying each corresponding weight in increments of 0.25. Specifically, we fix one of the terms (*e.g.*, $\mathcal{L}_{\text{dm}} = 0.00$) while tuning the other, and vice versa. As shown in Table 8, we empirically select $\lambda_{\text{dm}} = 0.50$ and $\lambda_\epsilon = 1.00$ based on the MUSIQ results.

### A.3 COMPREHENSIVE ANALYSIS OF $\mathcal{L}_\epsilon$

As shown in Table 6 of the main paper, the proposed $\mathcal{L}_\epsilon$ exhibits strong synergy with the other loss terms (*i.e.*, $\mathcal{L}_{\text{adv}}$ and $\mathcal{L}_{\text{dm}}$). In addition, Figure 4 of the main paper demonstrates that $\mathcal{L}_\epsilon$ enables the predicted noise $\epsilon_{\text{SR}}$ to align with the input LR image $\mathbf{I}_{\text{LR}}$.

Figure 6 further highlights the overall effectiveness of $\mathcal{L}_\epsilon$. As shown in Figure 6c, without $\mathcal{L}_\epsilon$, the predicted noise does not correspond to the input image, leading to input-irrelevant artifacts. Since the noise is otherwise unguided, the model tends to predict noise patterns that the pretrained denoiser $\mu_\psi$ finds easiest to denoise, rather than noise reflecting the input content. Figure 6d visualizes the distribution of $\epsilon_{\text{SR}}$. Here, because the pretrained denoiser $\mu_\psi$ is optimized to denoise Gaussian noise, $\epsilon_{\text{SR}}$ also follows a Gaussian distribution $\epsilon \sim \mathcal{N}(\mathbf{0}, \mathbf{I})$ even when $\mathcal{L}_\epsilon$ is not used. Therefore, to analyze meaningful differences, we plot the distribution at a fixed spatial location $(i, j)$, unlike Figure 4b of the main paper which shows the distribution of the whole noise tensor. As shown, without $\mathcal{L}_\epsilon$ (*i.e.*, red bar in the Figure 6d), the predicted values remain almost identical across different inputs, exhibiting small variance and little diversity. In contrast, with $\mathcal{L}_\epsilon$ (*i.e.*, blue bar in the Figure 6d), the distribution at the same spatial coordinate $(i, j)$ still follows a Gaussian and varies meaningfully according to the input image. This confirms that $\mathcal{L}_\epsilon$ guides the predicted noise to be input-aware, complementing the other loss terms, which operate on the image representation $\mathbf{Z}_{\text{SR}}$, in an orthogonal manner.

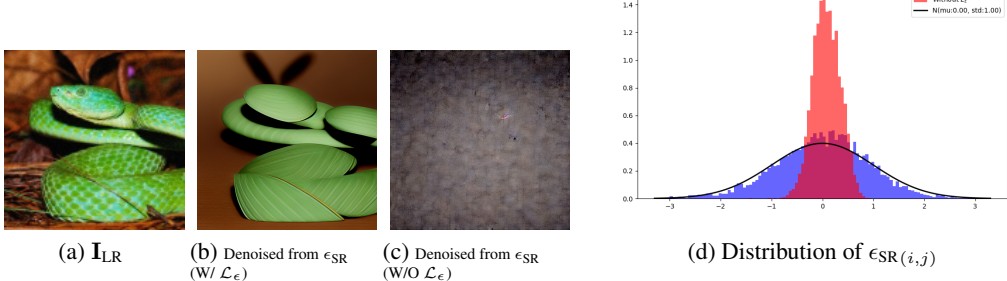

(a) $\mathbf{I}_{\text{LR}}$  (b) Denoised from $\epsilon_{\text{SR}}$ (W/ $\mathcal{L}_\epsilon$)  (c) Denoised from $\epsilon_{\text{SR}}$ (W/O $\mathcal{L}_\epsilon$)  (d) Distribution of $\epsilon_{\text{SR}(i,j)}$

Figure 6: **Analysis of the proposed $\mathcal{L}_\epsilon$.** (Left) Following the protocol in Table 4 of the main paper, the encoder transforms the leftmost LR inputs (a) into $\mathbf{Z}_{\text{SR}}$ using two different $\epsilon_{\text{SR}}$ (one from the model with $\mathcal{L}_\epsilon$ and one from the model without $\mathcal{L}_\epsilon$). When only $\mathcal{L}_\epsilon$ is applied, the diffusion process initialized with $\epsilon_{\text{SR}}$ generates outputs that remain aligned with the input content, as shown in (b) and (c). (Right) We visualize the distribution of the predicted noise $\epsilon_{\text{SR}}$ at a specific spatial position $(i, j)$. Without $\mathcal{L}_\epsilon$ (red bar), the predicted noise remains nearly identical across different inputs, indicating that the noise lacks semantic adaptation. In contrast, with $\mathcal{L}_\epsilon$ (blue bar), the predicted noise varies according to the input, demonstrating its content-aware behavior.

### A.4 COMPREHENSIVE ANALYSIS OF $\mathcal{L}_{\text{DM}}$

Similar to $\mathcal{L}_\epsilon$ in Section A.3, the proposed $\mathcal{L}_{\text{dm}}$ also exhibits strong synergy with the other loss terms, as shown in Table 6 of the main paper. Unlike the adversarial loss $\mathcal{L}_{\text{adv}}$, which encourages $\mathbf{Z}_{\text{SR}}$ to

move toward the overall distribution of real HR images, regardless of their semantic content, $\mathcal{L}_{\mathrm{dm}}$ specifically enforces distribution matching between samples of the same underlying content (*i.e.*, between $\mathbf{Z}_{\mathrm{SR}}$ and $\mathbf{x}_{\mathrm{SR}}^0$). Because of this property, $\mathcal{L}_{\mathrm{dm}}$ not only guides $\mathbf{Z}_{\mathrm{SR}}$ toward the real distribution (as $\mathcal{L}_{\mathrm{adv}}$ does) but also ensures that it aligns with the content-specific distribution found in the corresponding clean counterpart $\mathbf{x}_{\mathrm{SR}}^0$.

Table 9 further shows how $\mathcal{L}_{\mathrm{dm}}$ promote this content-consistent distribution alignment, going beyond what can be achieved with the less targeted adversarial loss. As shown, when $\mathcal{L}_{\mathrm{dm}}$ is used alongside $\mathcal{L}_{\mathrm{adv}}$, the model not only improves on non-reference perceptual metrics but also achieves gains on distortion-based measures. Although the improvements may seem small, they are consistent across both perceptual and distortion metrics, effectively mitigating the traditional distortion-perception trade-off, and therefore represent a meaningful enhancement to overall SR performance.

Table 9: $\times$ **4 SR quantitative comparison on RealSR dataset with various objective combinations.**

| $\mathcal{L}_{\mathrm{adv}}$ | $\mathcal{L}_{\mathrm{dm}}$ | PSNR↑ | SSIM↑ | LPIPS↓ | LIQE↑ | CLIPIQA↑ | TOPIQ (NR)↑ | NIMA↑ | MUSIQ↑ |
|---|---|---|---|---|---|---|---|---|---|
| ✓ | ✗ | 24.981 | 0.710 | 0.328 | 4.492 | 0.726 | 0.698 | 5.370 | 70.923 |
| ✓ | ✓ | 24.984 | 0.710 | 0.317 | 4.485 | 0.732 | 0.702 | 5.374 | 71.489 |

### A.5 DISTRIBUTION OF THE PREDICTED TIMESTEPS

Figure 7 shows the overall distribution of the predicted timesteps $\hat{t}$. As shown, the predicted timesteps(*i.e.*, blue lines in Figure 7) closely align with the normalized ground-truth timesteps (*i.e.*, red lines in Figure 7), achieving an ***RMSE of 7.74*** within the range $\in [0, 500]$. This demonstrates that our timestep estimator effectively predicts timesteps consistent with the distortion level of the LR input $\mathbf{I}_{\mathrm{LR}}$. For this visualization, we use the first 100 images from the ImageNet dataset to compare the predicted and ground-truth timesteps. For the rationale behind using the LPIPS score as the ground-truth timestep, please refer to Table 5 and Section 4.5.

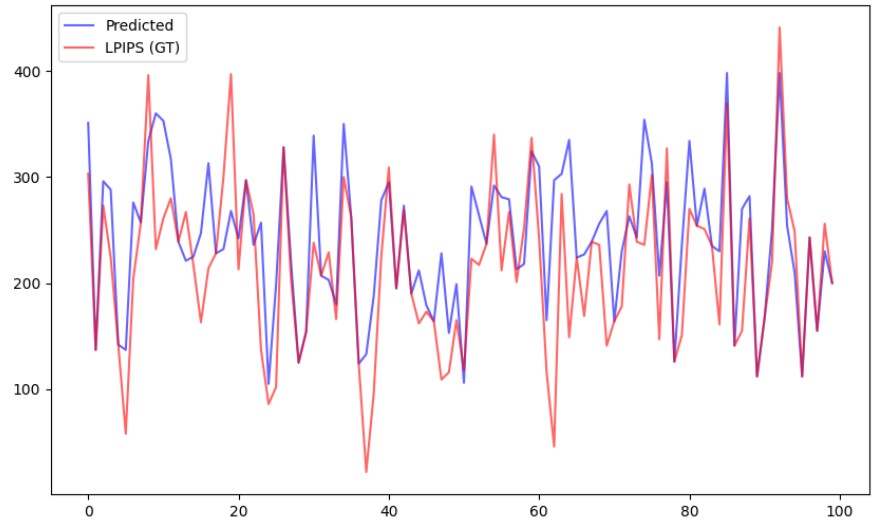

Figure 7: **Visualization of ground truth and predicted timesteps.**

### A.6 ADDITIONAL VISUAL COMPARISON

Figures 8, 9, and 10 present additional visual comparisons between our DM-SR and previous SR methods. Here, we present a comparison on real-world low-resolution images by collecting several

historical photographs from open-source resources. The results clearly demonstrate that DM-SR consistently generates perceptually superior outputs compared to existing approaches.

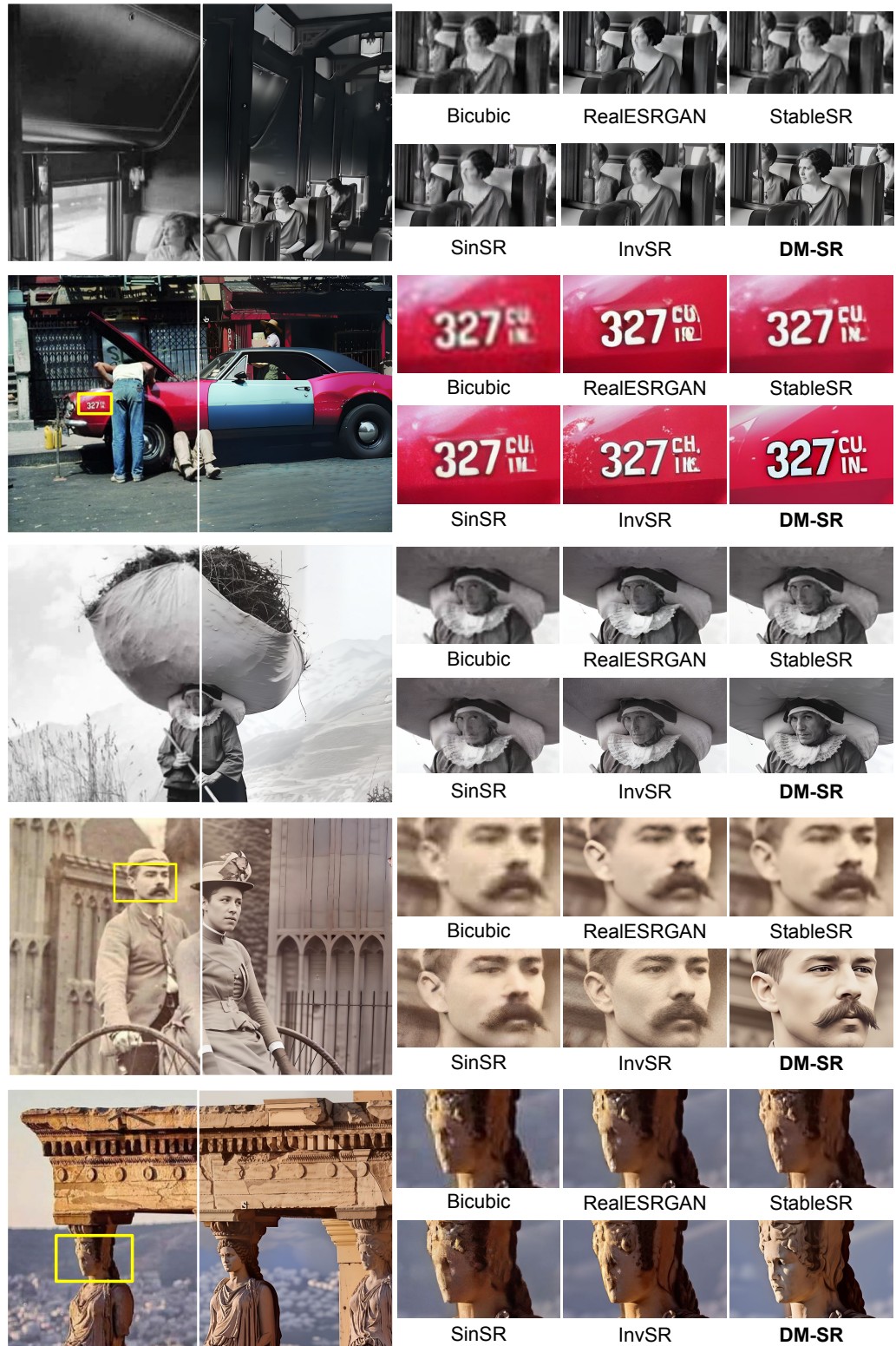

Figure 8: ×4 **super-resolution qualitative comparison on various real-world images.** (Zoom-in for best view)

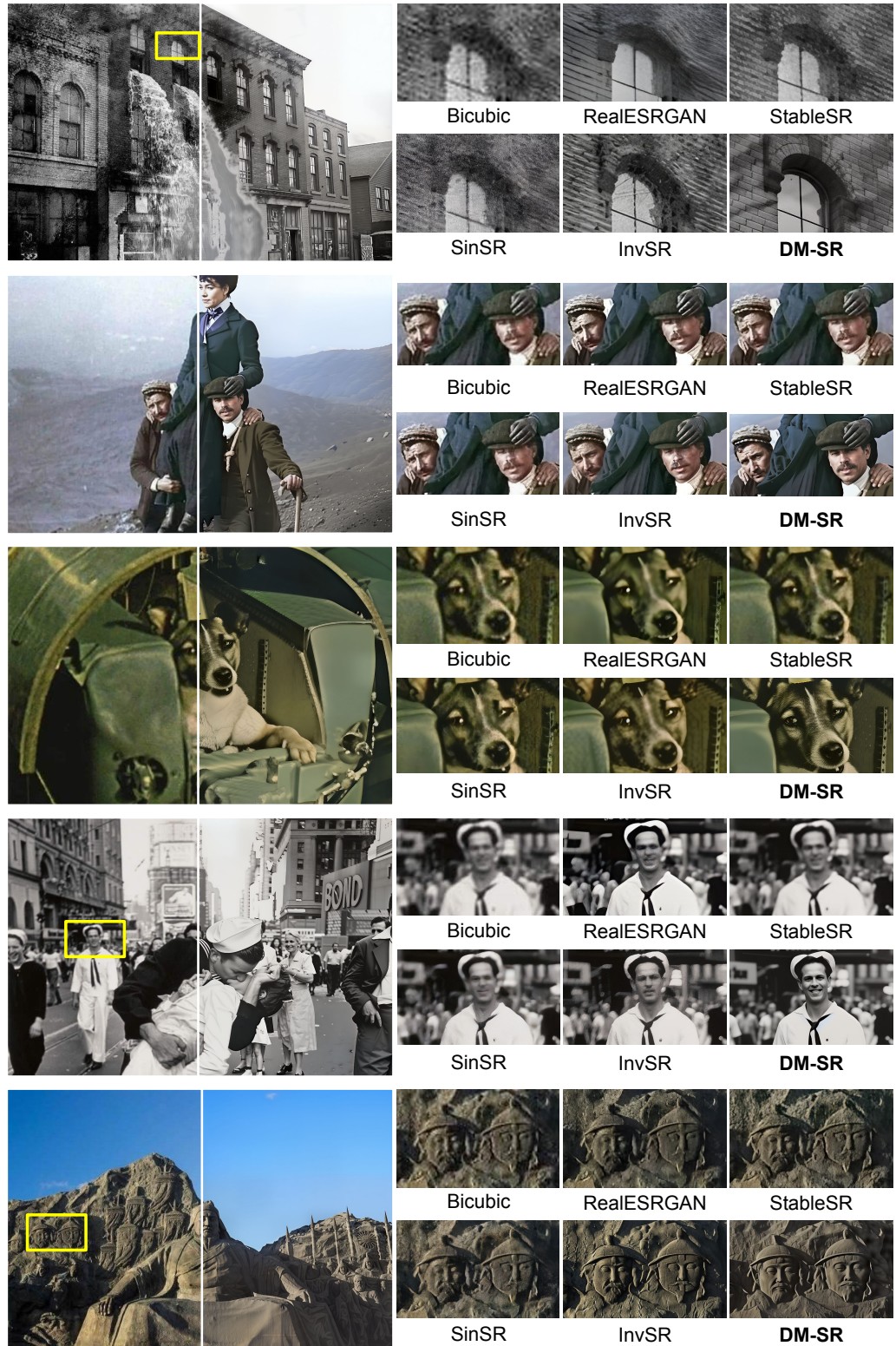

Figure 9: ×4 **super-resolution qualitative comparison on various real-world images.** (Zoom-in for best view)

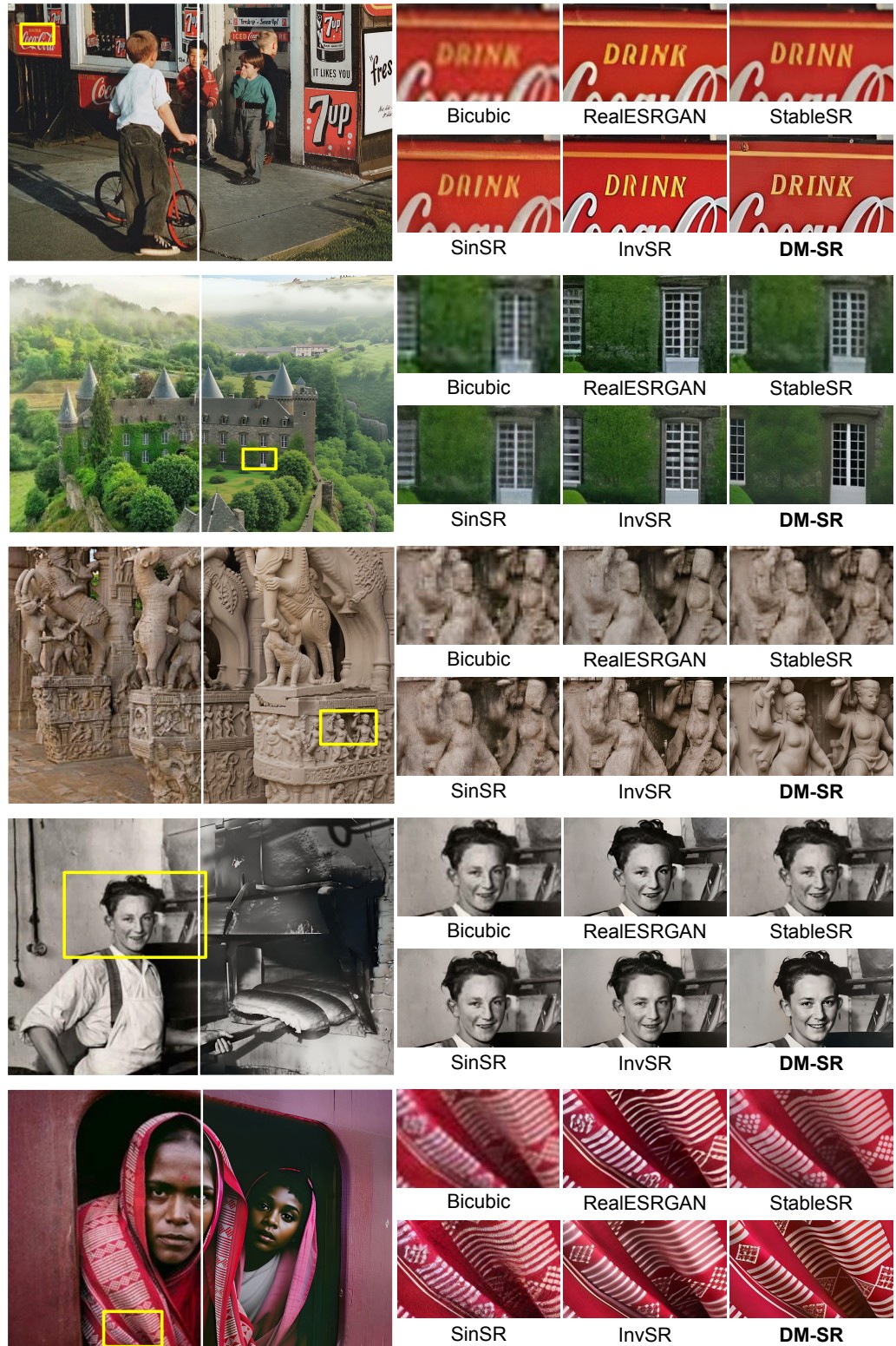

Figure 10: ×4 **super-resolution qualitative comparison on various real-world images.** (Zoom-in for best view)

