# OpenReview forum: "Bridging the Distribution Gap to Harness Pretrained Diffusion Priors for Super-Resolution"
_ICLR.cc/2026/Conference — ICLR 2026 Poster_

### Official Review · Reviewer_1GUs · 2025-10-25

**Soundness:** 3
**Presentation:** 2
**Contribution:** 3
**Rating:** 6
**Confidence:** 3

**Summary:**

This paper, "BRIDGING THE DISTRIBUTION GAP TO HARNESS PRE-TRAINED DIFFUSION PRIORS FOR SUPER-RESOLUTION," proposes DM-SR, a novel framework for single-image super-resolution (SR) that leverages the strong generative prior of a fixed, pretrained diffusion model (SD-Turbo is used). The core idea is to address the distribution mismatch between low-resolution (LR) inputs and the diffusion model's native training space (Gaussian-corrupted images) by training only an image encoder ($E_{\theta}$). This encoder adaptively maps the LR image into a timestep-dependent latent distribution, where the timestep $\hat{t}$ is predicted based on the input's degradation level. The resulting latent is decomposed into an image part ($Z_{SR}$) and a noise part ($\epsilon_{SR}$) using the fixed denoiser ($\mu_{\psi}$). The encoder is optimized using a comprehensive loss function that includes a novel conditional adversarial loss, a distribution matching loss, and a noise regularization loss. DM-SR achieves superior perceptual quality with a single, highly efficient step5

**Strengths:**

1. Preservation of Generative Prior: The core strength is that the method fully utilizes the fixed, pretrained denoiser ($\mu_{\psi}$), maintaining its powerful generative capabilities without compromise.
2. Efficiency: DM-SR is a single-step method, demonstrating the fastest runtime among competitive diffusion-based SR approaches while delivering outstanding perceptual results.
3. Noise-Adaptive Alignment: The dynamic prediction of a timestep ($\hat{t}$) based on the input image's degradation level is a clever mechanism to allocate the appropriate amount of generative power.

**Weaknesses:**

1. Dependency on Timestep Estimator: The success of the method critically depends on the accuracy of the Timestep Estimator ($T$), which is trained to predict a degradation level (normalized LPIPS). The justification for LPIPS as the "ground truth" for the degradation level could be elaborated more in the main paper。
2. Fixed Text Condition: The denoiser $\mu_{\psi}$ is conditioned using a fixed, generic text prompt ("High-quality, high-contrast, photo-realistic, clean..."). This design choice is simple but potentially limits the full capacity of the diffusion model, which is typically capable of content-aware generation via text conditions.

**Questions:**

1. Timestep Estimator Analysis: The reliance on the LPIPS score [0, 500] as the ground truth for $\hat{t}$ is novel but requires further analysis. Please show how the performance is affected if the estimated timestep $\hat{t}$ is replaced by a fixed, arbitrary timestep (e.g., $t=250$) across all samples. This would directly isolate the contribution of the adaptive alignment mechanism.
2. Multi-Step Comparison: The single-step generation is highly efficient. However, since $X_{SR}^{\hat{t}}$ is meant to align with the noisy latent $X_{HR}^{\tilde{t}}$, it is a valid input for a multi-step DDPM process. Please provide a comparison between DM-SR-1 (decoding $Z_{SR}$) and running a full multi-step sampler (e.g., 50 steps) initialized from $X_{SR}^{\hat{t}}$. This would confirm if the single-step is truly the optimal output.

---

> ### Author Response · Authors · 2025-11-13
>
> # (1) Dependency on Timestep Estimator.
> Because the timestep is used as an integer inside the UNet, it is difficult to learn purely from backpropagation without explicit supervision; direct supervision on the timestep is therefore necessary. While LPIPS may not be the theoretically optimal target, we empirically found that using the LPIPS score as the ground truth for the timestep works well. For variants, we also tried normalized, averaged pixel distances and SSIM. In an alternative approach, given an initial prediction $\hat{t}$, we super-resolve with $\hat{t}-1$ and $\hat{t}+1$ and then supervise $\hat{t}$ to match the timestep that minimizes
> $||Z_{\text{SR}} - X^{0}_{\text{HR}}||$. However, this yielded unstable gradients in our experiments and collapsed to a single timestep regardless of the input. Consequently, we adopt LPIPS as the supervision signal for timestep prediction. We have revised the Section 4.5 (L454–464) and added Table 5 to reflect this content.
>
> # (2) Fixed Text Condition.
> Table 7 (originally Table 6) and Section 4.7 (originally Section 4.6) in the main paper present the corresponding results, demonstrating that using input-specific text prompts further improves perceptual performance.
> However, because generating these prompts requires an additional vision–language model, this approach introduces extra latency, leading to a trade-off between efficiency and effectiveness.
>
>
> # (3) Timestep Estimator Analysis.
> Table 3 and Section 4.3 of the main paper present the corresponding results, demonstrating the validity of our timestep estimator $\mathcal{T}$.
> Additionally, as suggested by the reviewer, we report the results with the timestep fixed at 250 in Table 3.
>
>
> # (4) Multi-Step Comparison.
> Table 4 and Section 4.4 of the main paper present results analyzing the impact of different sampling step numbers in our DM-SR framework.
> Although a single step is not optimal for every metric, it provides the best balance between performance and efficiency.
> As discussed in Section 4.4, when the number of sampling steps exceeds 20 (e.g., 50), the performance degrades significantly.

---

> > ### Comment · Reviewer_1GUs · 2025-11-24
> >
> > Thanks for your response.
> >
> > About multi-step comparison, I wonder how you determine the timesteps for multi-step inference? Assume that the Timestep estimator makes an estimation with t=432, for the case the 4-step inference, what would be the intermediate inference timesteps in your experiment?
> >
> > In another case, if the Timestep estimator makes another estimation with t=382, again, if we also conduct 4-step inference, would the intermediate timesteps be different?
> >
> > Finally, I would like to see the distribution of timesteps predicted by your Timestep Estimator on the test set.

---

> ### Author Response · Authors · 2025-11-24
>
> We sincerely thank reviewer 1GUs for the response.
> Regarding the distribution of timesteps, we provide a detailed visualization of both the ground-truth and predicted timesteps in Figure 7 and Section A.5 of the Appendix.
> These results demonstrate that our timestep estimator successfully predicts timesteps consistent with the degradation level (LPIPS), achieving an RMSE of 7.74 between ground-truth and predicted values.
>
> For multi-step sampling, the subsequent timesteps are scheduled linearly based on the initially predicted timestep.
> For example, if the initially predicted timestep is $\hat{t} = 382$ and the number of sampling steps is 4, the remaining timesteps are set to $[(3\hat{t})/4, (2\hat{t})/4, (1\hat{t})/4] = [287, 191, 96]$.
>
> An alternative strategy would be to re-estimate the timestep at each iteration by replacing the input $I_{LR}$ with the intermediate super-resolved image $I_{SR}$ from the previous step.
> However, we find that the predicted timestep from $I_{SR}$ tends to be excessively small, resulting in minimal updates across iterations and making multi-step sampling behave almost identically to single-step sampling.
>
> To ensure clarity, we have also revised the presentation in Section 4.4 (Lines 424–427 in the main paper) to include these details more explicitly.

---

> > ### Comment · Reviewer_1GUs · 2025-11-27
> >
> > Thank you for your response. I will maintain my current recommendation.

---

> > > ### Author Response · Authors · 2025-11-27
> > >
> > > We sincerely thank reviewer 1GUs for the constructive discussion. We remain fully open to further discussion and would be glad to address any additional questions or concerns.

---

### Official Review · Reviewer_4wwB · 2025-10-30

**Soundness:** 3
**Presentation:** 3
**Contribution:** 3
**Rating:** 6
**Confidence:** 3

**Summary:**

The paper introduces a novel framework for single-image super-resolution that leverages pretrained diffusion models without requiring their fine-tuning or iterative denoising steps. DM-SR trains an image encoder to map LR inputs into a latent distribution aligned with the diffusion model’s training space. The method adaptively predicts the appropriate noise level for each input, ensuring optimal alignment with the diffusion model’s timestep-dependent distribution. Extensive experiments on synthetic and real-world benchmarks demonstrate that DM-SR achieves state-of-the-art perceptual results.

**Strengths:**

1. This paper proposes a novel way to leverage pre-trained diffusion model for SR, by aligning the latent representations of a LR image with an intermediate noisy latent representation of diffusion model. It does not require training the pre-trained model which has the merits of take full advantage of the pre-trained knowledge.
2. The paper is well written, and the work and contribution are clearly presented.
3. Extensive experiments are conducted showing the superior non-reference performance compared to existing pre-trained diffusion based method.

**Weaknesses:**

1. It would strengthen the paper to include a more in-depth analysis comparing the proposed model with single-step diffusion-based super-resolution methods such as invSR. Since both approaches share similar motivations and framework designs, a detailed explanation of the differences—and why the proposed method achieves superior results—would help clarify its unique contributions and advantages.

2. Most of the comparison methods in the experiments are single-step diffusion-based super-resolution approaches. However, there are other ways to leverage pretrained diffusion models for super-resolution. Including comparisons with a broader range of diffusion-based SR methods would provide a more comprehensive understanding of how different strategies for utilizing pretrained diffusion models impact SR performance.

3. The proposed method does not demonstrate strong performance on reference-based metrics, and in some cases, it underperforms compared to certain single-step diffusion-based super-resolution approaches. Providing a more detailed explanation for these results would be very helpful in understanding the limitations and potential areas for improvement.

**Questions:**

Please responds to my concerns in Weaknesses.

---

> ### Author Response · Authors · 2025-11-13
>
> # (1) Comparison with InvSR.
> We thank Reviewer 4wwB for the insightful feedback.
> Although DM-SR shares a similar motivation with InvSR, our approach differs significantly in its underlying design and methodology.
>
> ## Assumption about noisy LR vs. noisy HR.
> First, InvSR treats noisy LR and noisy HR as indistinguishable and, based on this premise, forms the UNet input by combining the predicted noise with the latent of the LR image. In strict terms, this assumption does not hold: adding noise preserves a fixed proportion of the underlying signal, so the resulting noisy LR and noisy HR distributions are not identical.
> Specifically, when timestep is set to small, noisy version of LR and HR images may retain original input image, resulting in more server assumption error. In contrast, the encoder in DM-SR learns to map the LR image to its corresponding noisy HR representation.
>
> ## Supervision of the predicted noise.
> Second, InvSR predicts noise without direct supervision. In contrast, DM-SR introduces an explicit supervision term $L_{\epsilon}$ for the predicted noise. This direct constraint yields measurable gains in super-resolution performance, as shown in Table 6 of the main paper.
>
> ## Timestep selection strategy.
> Third, InvSR uses a fixed timestep for all LR inputs. On the other hand, DM-SR predicts the timestep adaptively according to the degradation level of each input, enabling the denoising schedule to better match the data. The benefit of this adaptive timestep prediction is clearly demonstrated in Table 3 of the main paper.
>
> Furthermore, we have revised the introduction (Lines 73–77 in the main paper) to carefully incorporate the corresponding content.
>
>
> # (2) Broader Comparison.
> As recent diffusion-based methods have adopted single-step generation and achieved improved results through advanced techniques, we primarily compare our method with recent single-step SR approaches.
> Nevertheless, we fully agree with the reviewer’s suggestion.
> The table below includes comparisons on **DRealSR** and **RealSR**, respectively, with a broader range of diffusion-based SR methods, further demonstrating the effectiveness of our DM-SR.
>
> | Method         | CLIPIQA ↑ | MANIQA ↑ | MUSIQ ↑ |
> |----------------|:----------:|:---------:|:--------:|
> | StableSR-200   | 0.6356 | 0.5601 | 58.51 |
> | DiffBIR-50     | 0.6395 | 0.5930 | 61.07 |
> | SeeSR-50       | 0.6804 | 0.6042 | 64.93 |
> | PASD-20        | 0.6808 | 0.6169 | 64.87 |
> | ResShift-15    | 0.5342 | 0.4586 | 50.60 |
> | **DM-SR-1 (Ours)** | **0.7530** | **0.5880** | **69.04** |
>
>
> | Method         | CLIPIQA ↑ | MANIQA ↑ | MUSIQ ↑ |
> |----------------|:----------:|:---------:|:--------:|
> | StableSR-200   | 0.6178 | 0.6221 | 65.78 |
> | DiffBIR-50     | 0.6463 | 0.6246 | 64.98 |
> | SeeSR-50       | 0.6612 | 0.6442 | 69.77 |
> | PASD-20        | 0.6620 | 0.6487 | 68.75 |
> | ResShift-15    | 0.5444 | 0.5285 | 58.43 |
> | **DM-SR (Ours)** | **0.7320** | **0.6010** | **71.49** |
>
>
> # (3) Performance on Reference-based Metrics.
> As discussed in Section 4.2, there exists a standard trade-off between perceptual quality and distortion.
> We fully agree that further investigation of this issue is important. Accordingly, we have moved the Limitation section from Appendix to Table 5 of the main paper and provided a more detailed explanation.
> As shown in Figure 5 (f), although our DM-SR achieves higher perceptual scores than prior methods Figure 5 (b)–(e), it introduces noticeable distortion, such as a change in pupil color.
> To mitigate this, we adopt input-specific text prompts that describe attributes of the LR image (e.g., specifying that the pupil is light gray), which produces the gray-colored pupil in Figure 5 (g).
> We hope this strategy addresses the reviewer’s concern and represents a promising direction for future work, enabling a better balance between perceptual realism and content fidelity through adaptive text guidance.

---

### Official Review · Reviewer_85hy · 2025-10-30

**Soundness:** 3
**Presentation:** 2
**Contribution:** 3
**Rating:** 6
**Confidence:** 3

**Summary:**

This paper tackles the "distribution gap" that arises when applying pre-trained diffusion models (DMs) to image super-resolution (SR). The authors note that DMs are trained on Gaussian-corrupted images, a distribution that low-resolution (LR) inputs do not match. Existing methods attempt to solve this by fine-tuning the DM, which can weaken its powerful generative prior, or by using multi-step inference, which is computationally expensive. The authors propose DM-SR, a novel framework that keeps the pre-trained diffusion model completely frozen. Instead, it trains only an image encoder. This encoder is designed to do two things: (1) map the LR input image into the latent distribution based on the frozen DM, and (2) adaptively predict the appropriate noise level (timestep) based on the degradation of the input image . This approach allows the framework to harness the full generative power of the pre-trained model and achieve state-of-the-art perceptual quality in a single inference step. Extensive experiments show the method is highly efficient (achieving the fastest runtime among diffusion-based methods ) and effective on both synthetic and real-world datasets.

**Strengths:**

The prevailing paradigms for diffusion-based SR involve either fine-tuning the denoiser (which compromises the prior ) or complex multi-step conditioning. The proposed idea of keeping the denoiser frozen and instead learning to map the input into its native distribution is an elegant and powerful alternative. This approach simultaneously solves two major problems: it preserves the integrity of the powerful generative prior and enables extremely efficient single-step inference. This is a valuable conceptual contribution that will likely be influential for the ICLR community and future work in diffusion-based image restoration.

**Weaknesses:**

**Major Weaknesses:**

(1) My primary concern is the complexity of the objective function. The final loss is a combination of five different loss terms, which seems to be very complex. The weighting parameters are simply stated as "empirically chosen", and the paper provides no clear explanation or justification for why these specific hyperparameters were set. This makes the method seem brittle and difficult to reproduce. While the ablation in Table 5 shows that all components contribute to the final result 3, the paper lacks a strong justification for why this specific combination is necessary, especially for the two loss terms.

(2) The paper acknowledges the method's poor performance on distortion metrics (PSNR/SSIM) and frames it as a standard perceptual-distortion trade-off. However, the limitation shown in Figure 5 (changing the eye's pupil color from gray to black ) is a clear example of sacrificing semantic fidelity for perceptual realism. This is a significant issue. While the black pupil may appear more "realistic" (which is why the NR metrics are high), it is an uncontrolled hallucination—the model is generating a feature not present in the input. There is no evidence that this "fix" is semantically correct, and no guarantee that it can be correctly generated every time. This lack of faithfulness deserves a more in-depth discussion.

**Minor Weaknesses:**

(1) The analysis in Table 4 regarding the number of steps is confusing. While the paper explains that too many steps (10+) hurt performance due to the SD-Turbo backbone, it fails to explain the trend for 1, 2, and 5 steps. For some metrics (like LIQE), performance peaks at 2 or 5 steps and is slightly worse at 1 step. The paper does not address this fluctuation, making it seem like random noise rather than a significant, well-understood trend.

**Questions:**

(1) Could you provide more justification for using the LPIPS score as the regression target for the timestep estimator? Why was this metric chosen over other, simpler metrics (like PSNR) or a learned degradation classifier?

(2) In Table 4, the model performance varies with the number of steps. The paper explains why many steps (10+) hurt performance due to the SD-Turbo backbone. However, it does not explain the trend for 1, 2, and 5 steps, where performance seems to peak at 2 or 5 steps and then slightly decrease at 1 step (e.g., for LIQE ). Is this slight variation statistically significant, or is it more likely a random fluctuation within a stable performance range for low step counts?

---

> ### Author Response · Authors · 2025-11-13
>
> # (1) Hyperparameter.
>  We thank Reviewer 85hy for the insightful feedback. For the conventional loss terms, including $L_{\text{L1}}$, $L_{\text{per}}$, and $L_{\text{adv}}$, we follow setting in InvSR. For the newly introduced losses, $L_{\text{dm}}$ and $L_{\epsilon}$, we conduct ablation studies by varying each corresponding weight in increments of $0.25$.
> Specifically, we fix one of the terms (e.g., $\mathcal{L}_{\text{dm}}=0.00$) while tuning the other, and vice versa.
> Please refer to the Appendix A.2 for the details.
>
> # (2) In-depth Discussion on Hallucination.
>  As the reviewer pointed out, the perceptual–distortion trade-off remains a non-negligible issue.
> Accordingly, we have moved the Limitation section (originally placed in the Appendix) into Figure 5 of the main paper to discuss potential strategies for mitigating content distortion.
> Specifically, we adopt customized, input-specific text prompts that specify attributes of the LR image (e.g., indicating that the pupil is light-colored gray), which results in the gray-colored pupil shown in Figure 5 (g).
> While this approach may not fully resolve the reviewer’s concern, we believe it highlights an important direction for future research on leveraging input-specific text guidance to better balance perceptual quality and content fidelity.
>
>
>
> # (3) Trend for 1, 2, and 5 Steps.
> Specifically, SD-Turbo is trained using a distillation strategy, where longer sampling trajectories (i.e., a large number of steps) lead to excessive error accumulation and degraded performance.
> Since SD-Turbo achieves its best results with a small number of sampling steps (typically 1–5 are all fine), our SR performance also remains optimal within this range.
> We have revised the Section 4.4 (L428) to reflect this content.
>
>
> # (4) Ground Truth for the Timestep.
> Because the timestep is used as an integer inside the UNet, it is difficult to learn purely from backpropagation without explicit supervision; direct supervision on the timestep is therefore necessary. While LPIPS may not be the theoretically optimal target, we empirically found that using the LPIPS score as the ground truth for the timestep works well. For variants, we also tried normalized, averaged pixel distances and SSIM. In an alternative approach, given an initial prediction $\hat{t}$, we super-resolve with $\hat{t}-1$ and $\hat{t}+1$ and then supervise $\hat{t}$ to match the timestep that minimizes
> $||Z_{\text{SR}} - X^{0}_{\text{HR}}||$. However, this yielded unstable gradients in our experiments and collapsed to a single timestep regardless of the input. Consequently, we adopt LPIPS as the supervision signal for timestep prediction. We have revised the Section 4.5 (L454–464) and added Table 5 to reflect this content.

---

### Official Review · Reviewer_KAnJ · 2025-10-31

**Soundness:** 2
**Presentation:** 2
**Contribution:** 3
**Rating:** 2
**Confidence:** 5

**Summary:**

This paper proposes DM-SR, a diffusion-based super-resolution (SR) framework designed to bridge the distribution mismatch between low-resolution (LR) images and the Gaussian-corrupted image space used by pretrained diffusion models.
Instead of fine-tuning the diffusion model (which may degrade its generative prior), the authors train a lightweight image encoder that projects LR inputs into a diffusion-aligned latent distribution while keeping the pretrained denoiser fixed.

**Strengths:**

This paper proposes a diffusion-based SR method without the requirement of modifying the pre-trained Diffusion model weights, and achieves promising results.

**Weaknesses:**

Q1. The second paragraph identifies the limitations of existing diffusion-based SR methods, such as sampling from pure Gaussian noise and the corruption of generative priors during fine-tuning. This paper aims to address these issues. However, similar challenges have already been discussed in several prior works, even though they have not been completely resolved. Therefore, it is essential for the authors to analyze and discuss these earlier studies to properly position their contribution and demonstrate respect for prior research efforts. Without such discussion, the motivation appears insufficiently justified.

Q2. The paper states that the ground truth for the timestep predictor is obtained using the LPIPS metric. More technical details about this process are needed — for example, how LPIPS is used to determine the optimal timestep.

Q3. The proposed method employs a combination of multiple loss terms. From Table 5, it seems that the performance gain mainly originates from the adversarial loss $L_{adv}$. Given the adversarial loss $L_{adv}$, the necessity of the other two loss components of $L_{dm}$ and $L_{eps}$ should be further validated.

Q4. The reference metric values for the ImageNet dataset should be reported for completeness. Additionally, the synthesis process used to construct the ImageNet dataset for the experiments should be clearly described, including degradation types, resolution settings, and data splits. This information is crucial for ensuring experimental transparency and reproducibility.

Q5. Since this work closely follows the recent InvSR framework, it would be beneficial to include a direct comparison with InvSR in Table 4.  It seems that the authors deliberately weakened the relation of the proposed method to InvSR, as pointed out in Q1.

**Questions:**

NA

---

> ### Author Response · Authors · 2025-11-13
>
> # (1) Analysis on Earlier Studies.
> We thank Reviewer KAnJ for the insightful feedback. As far as we are aware, aside from our DM-SR and InvSR, most recent diffusion-based SR methods focus on updating the UNet (including via LoRA) without explicitly seeking the optimal noise for denoising. Although DM-SR and InvSR both leverage predicted noise, our approach differs substantively from InvSR.
>
> ## Assumption about noisy LR vs. noisy HR.
> First, InvSR treats noisy LR and noisy HR as indistinguishable and, based on this premise, forms the UNet input by combining the predicted noise with the latent of the LR image. In strict terms, this assumption does not hold: adding noise preserves a fixed proportion of the underlying signal (i.e., either LR or HR), so the resulting noisy LR and noisy HR distributions are not identical.
> Specifically, when timestep is set to small, noisy version of LR and HR images may retain original input image, resulting in more server assumption error.
> In contrast, the encoder in DM-SR learns to map the LR image to its corresponding noisy HR representation.
>
> ## Supervision of the predicted noise.
> Second, InvSR predicts noise without direct supervision. In contrast, DM-SR introduces an explicit supervision term $\mathcal{L}_{\epsilon}$ for the predicted noise. This direct constraint yields measurable gains in super-resolution performance, as shown in Table 6 of the main paper.
>
> ## Timestep selection strategy.
> Third, InvSR uses a fixed timestep for all LR inputs. On the other hand, DM-SR predicts the timestep adaptively according to the degradation level of each input, enabling the denoising schedule to better match the data. The benefit of this adaptive timestep prediction is clearly demonstrated in Table 3 of the main paper.
>
> Furthermore, we have revised the introduction (Lines 73–77 in the main paper) to carefully incorporate the corresponding content.
>
>
> # (2) Ground Truth for Timestep.
> Because the timestep is used as an integer inside the UNet, it is difficult to learn purely from backpropagation without explicit supervision; direct supervision on the timestep is therefore necessary. While LPIPS may not be the theoretically optimal target, we empirically found that using the LPIPS score as the ground truth for the timestep works well. For variants, we also tried normalized, averaged pixel distances and SSIM. In an alternative approach, given an initial prediction $\hat{t}$, we super-resolve with $\hat{t}-1$ and $\hat{t}+1$ and then supervise $\hat{t}$ to match the timestep that minimizes
> $||Z_{\text{SR}} - X^{0}_{\text{HR}}||$. However, this yielded unstable gradients in our experiments and collapsed to a single timestep regardless of the input. Consequently, we adopt LPIPS as the supervision signal for timestep prediction. We have revised the Section 4.5 (L454–464) and added Table 5 to reflect this content.
>
> # (3) Effect of the Loss Functions.
> Using all loss terms together substantially improves SR performance over using only the adversarial loss, showing strong synergy. Each component contributes complementary benefits beyond the baseline (i.e., without all three losses), suggesting that they handle largely independent factors and leave headroom for the others to further improve results. Moreover, the table below isolates the effects of $L_{\text{dm}}$ and $L_{\epsilon}$, when the adversarial loss is present; both terms provide additional gains.
>
> | Loss Setting          | LIQE ↑  | CLIPIQA ↑ | TOPIQ (NR) ↑ | NIMA ↑  | MUSIQ ↑  |
> |------------------------|:-----:|:-------:|:-----------:|:-----:|:------:|
> | $L_{adv}$                     | 4.579 | 0.779 | 0.694 | 5.480 | 69.947 |
> | $ L_{adv} $ $ L_{dm} $ | 4.510 | 0.781 | 0.694 | 5.481 | 70.146 |
> | $ L_{adv} $  $ L_{\epsilon} $ | 4.596 | 0.785 | 0.693 | 5.513 | 70.069 |
> | **All three losses (Ours)**         | **4.652** | **0.797** | **0.707** | **5.575** | **70.616** |

---

> ### Author Response · Authors · 2025-11-13
>
> # (4) ImageNet Protocol and Results.
> We follow the InvSR framework to generate LR images and use the degraded ImageNet dataset provided from them. We also report reference-based metrics on ImageNet in the table below. Similar to our RealSR results in Tables 1 and 2 of the main paper, our method does not attain the very best reference-based scores but remains competitive. We have revised the Section 4.1 (L308–309) to reflect this content.
>
>
> | Method              | PSNR ↑  | SSIM ↑  | LPIPS ↓ |
> |----------------------|:-------:|:-------:|:-------:|
> | RealESRGAN           | 24.768  | 0.724   | 0.230   |
> | StableSR-50          | 24.111  | 0.702   | 0.231   |
> | SinSR-1              | 24.923  | 0.703   | 0.220   |
> | OSEDiff-1            | 23.052  | 0.666   | 0.243   |
> | *InvSR-1*            | 22.007  | 0.647   | 0.255   |
> | DM-SR-1 (Ours)   | 22.037 | 0.653 | 0.249 |
>
>
> # (5) Direct comparison with InvSR.
> The table below compares DM-SR and InvSR across a range of sampling steps. DM-SR consistently outperforms InvSR under different sampling step number configurations.
> Here, for InvSR, we follow its original protocol when using 1–5 timesteps:
> [200], [200, 100], [200, 100, 50], [200, 150, 100, 50], and [250, 200, 150, 100, 50], respectively.
> For timestep numbers greater than 5, we generate linearly spaced timesteps from 250 to 0.
> Since InvSR is trained to predict the initial noise only at specific timesteps, it cannot accurately estimate noise for arbitrary intermediate steps; instead, it resorts to random noise sampling, leading to degraded performance when multiple steps are applied.
> In contrast, DM-SR does not assume a fixed correspondence between noise and specific timesteps, enabling more stable multi-step performance.
> For a analytic comparison between the two methods, please refer to (1) in our response.
>
> | # of Steps | Method | LIQE ↑ | CLIPIQA ↑ | TOPIQ (NR) ↑ | NIMA ↑ | MUSIQ ↑ |
> |:-----------:|:--------|:------:|:----------:|:--------------:|:------:|:-------:|
> | **20** | *InvSR* | 3.692 | 0.671 | 0.578 | 5.265 | 65.707 |
> |            | **DM-SR** | **4.152** | **0.729** | **0.737** | **5.081** | **67.800** |
> | **10** | *InvSR* | 3.817 | 0.680 | 0.590 | 5.287 | 66.420 |
> |            | **DM-SR** | **4.503** | **0.762** | **0.743** | **5.531** | **68.135** |
> | **5** | *InvSR* | 3.965 | 0.695 | 0.599 | 5.337 | 67.321 |
> |            | **DM-SR** | **4.773** | **0.768** | **0.749** | **5.756** | **69.948** |
> | **2** | *InvSR* | 4.192 | 0.713 | 0.618 | 5.405 | 68.975 |
> |            | **DM-SR** | **4.799** | **0.778** | **0.744** | **5.692** | **69.435** |
> | **1** | *InvSR* | 4.291 | 0.727 | 0.623 | 5.549 | 69.798 |
> |            | **DM-SR** | **4.652** | **0.797** | **0.707** | **5.575** | **70.616** |

---

> > ### Comment · Reviewer_KAnJ · 2025-11-15
> > **Feedback to author rebuttal**
> >
> > Thanks for the author's rapid response. At the first round review, my main concerns (Q1 and Q2) focused on the paper writing, which has been addressed in the revised manuscript. I thus increase my rating.
> >
> > As for Q3 on loss analysis, I'm still confused about the necessity of $L_{dm}$ and $L_{\epsilon}$. For comparison convenience, I summarize the performance gain brought by them:
> >
> > $L_{dm}$:  -0.069 (LIQE)  0.002 (CLIPIQA) 0.000 (TOPIQ) 0.001 (NIMA) 0.199 (MUSIQ);
> >
> > $L_{\epsilon}:~$   +0.017 (LIQE) 0.006 (CLIPIQA) -0.001 (TOPIQ)  0.033 (NIMA) 0.122.
> >
> > Honestly, such a slight performance gain can't convince me.

---

> > > ### Author Response · Authors · 2025-11-17
> > >
> > > We sincerely thank reviewer KAnJ for the prompt and thoughtful response.
> > > We are pleased to hear that the reviewer’s primary concerns regarding Q1 and Q2 have been sufficiently addressed.
> > >
> > > Regarding Q3, we appreciate the insightful comments and would like to further clarify the roles of each loss term, as well as how they complement one another.
> > >
> > > First, the adversarial loss $L_{adv}$ trains the SR network (i.e., the encoder in our DM-SR) to generate outputs that the discriminator perceives as real.
> > > However, adversarial learning groups all real and fake images together without considering differences in semantic or structural properties across samples.
> > > As a result, $L_{adv}$ may fail to provide meaningful gradients in certain cases: for example, gradients encouraging a **fake tree to resemble real water** are not semantically useful and may mislead the network.
> > >
> > > In contrast, $L_{dm}$ and $L_{\epsilon}$ are content-aware and enforce distribution matching between samples of the same underlying content.
> > > Specifically, $L_{dm}$ aligns the score functions between the corresponding contents (i.e., $X_{SR}^{\hat{t}}$ and $Z_{SR}$), ensuring that the distribution of a **fake tree is guided toward the distribution of its real tree counterpart** rather than any arbitrary real sample.
> > > $L_{\epsilon}$ aligns the predicted noise $\epsilon_{SR}$ with the semantic properties of the input image (as shown in Figure 4 and Section 4.8 of the main paper).
> > > This, in turn, influences both $X_{SR}^\hat{t}$ and $Z_{SR}$.
> > > Without $L_{\epsilon}$, the predicted noise becomes unregularized and may collapse to an input-agnostic noise pattern (what we observe in our experiments).

---

> > > > ### Comment · Reviewer_KAnJ · 2025-11-22
> > > > **Response Regarding Q3**
> > > >
> > > > The authors have demonstrated in Figure 4 that the optimized noise exhibits semantic consistency with the Low-quality image and obeys a Normal Gaussian distribution. However, there is no evidence showing that such a phenomenon is owing to the loss component of $L_{dm}$ and $L_{\epsilon}$.  Thus, it is still necessary to replot Fig. 4 after removing $L_{dm}$ and $L_{\epsilon}$ to support your claim.

---

> > > > > ### Author Response · Authors · 2025-11-24
> > > > >
> > > > > We sincerely thank reviewer KAnJ for the thoughtful and constructive feedback, which has greatly helped strengthen our work.
> > > > > In response to the reviewer’s suggestions, we provide additional analyses in Appendix A.3 and Appendix A.4.
> > > > >
> > > > > We would also like to clarify that the semantic consistency observed in the noise component (shown in Figure 4 of the main paper) is primarily attributed to $L_{\epsilon}$, rather than $L_{dm}$.
> > > > > To highlight this distinction, we include a replot of Figure 4 with $L_{\epsilon}$ removed in Figure 6 and further elaborate on this in Appendix A.3.
> > > > > Additionally, to demonstrate the semantic alignment induced by $L_{dm}$, we report distortion-based metrics in Table 9 and discuss them in Appendix A.4.
> > > > > We note that these metrics are evaluated on the RealSR dataset, where ground-truth HR images are available.
> > > > >
> > > > > We hope these clarifications and additional analyses address the reviewer’s concerns and further substantiate the contributions of our work.

---

> > > > > > ### Comment · Reviewer_KAnJ · 2025-11-24
> > > > > > **Response to author rebuttal**
> > > > > >
> > > > > > The additional results in the appendix have addressed my concern. I further improve my rating.

---

> > > > > > > ### Author Response · Authors · 2025-11-24
> > > > > > >
> > > > > > > We sincerely thank reviewer KAnJ once again for the prompt response and for providing constructive feedback that helped us further improve the clarity and quality of our paper.
> > > > > > > We remain fully open to additional discussion and will gladly address any remaining concerns.
> > > > > > > The reviewer KAnJ's insights are greatly appreciated, and we will do our best to incorporate any further suggestions to strengthen the work.

---

### Author Response · Authors · 2025-11-13
**Response to Reviewers**

We sincerely thank all reviewers for their valuable time, thoughtful feedback, and constructive comments.

We would like to note that Tables 5 and 6 of the original paper have been renumbered to Tables 7 and 8 due to the addition of a new table.
We deeply appreciate reviewers KAnJ, 85hy, 4wwB, and 1GUs for recognizing the contributions and soundness of our work.
We have made every effort to improve the presentation of the paper through careful revisions and additional explanations.
In the revised version of the paper, improvements for clearer presentation are marked in **red**, while revisions addressing the reviewers’ comments are marked in **blue**.

We hope that our responses and the updated manuscript adequately address the reviewers’ concerns, and we would be delighted to engage in further discussion.

---

### Author Response · Authors · 2025-11-28
**Dear ACs and Reviewers**

First of all, we sincerely thank ACs and reviewers for their time and effort. We have tried our best to address the concerns raised.

We would also like to clarify that, following our constructive discussions, all reviewers had set their scores to 6, 6, 6, and 6 as of November 24. Please refer to our detailed comments for further details. We truly appreciate everyone’s insightful feedback once again.

---

### Author Response · Authors · 2025-12-01
**Summary of Rebuttal and Discussions**

**Reviewer KAnJ (2 → 6):** Reviewer KAnJ’s primary concerns were the distinction between our method and prior approach (i.e., invSR), the justification for using LPIPS, and the effects of the proposed loss functions.
We addressed these concerns by clarifying our methodology in the revised manuscript and by providing ablation studies in Tables 5 and 9 of the paper. These resolved the reviewer’s concerns, leading to an increased rating from 2 to 6.

**Reviewer 85hy: (6):** Reviewer 85hy's primary concern was the selection of hyperparameters for the used loss terms. To address this, we included ablation studies with various hyperparameter configurations in Appendix A.2.

**Reviewer 4wwB: (6):** Reviewer 4wwB's primary concerns were the distinction between ours and prior approach and broader comparisons. We responded by revising the manuscript (Lines 73–77) and by expanding our comparisons accordingly.

**Reviewer 1GUs: (6 → 6):** Reviewer 1GUs’s primary concerns were the justification for using LPIPS and the specifics of our multi-step sampling design. In response, we provided additional ablation studies (Table 5) and improved presentation in Lines 424–427. After these clarifications, the reviewer maintained the initial positive rating.

---

### Meta-Review · Area_Chair_cQjj · 2026-01-06

**Summary:**

This work proposes a diffusion-based SR method that mitigates the mismatch between low-resolution images and the Gaussian noise space assumed by pretrained diffusion models. The proposed method does not require any training, preserves the original generative prior, and achieves strong perceptual quality with single-step inference. The paper received an initial average rating of 5.0 (2, 6, 6, 6). Reviewers raised concerns regarding discussion of prior studies (KAnJ, 4wwB, 1GUs), the effectiveness of LPIPS ground-truth for the timestep predictor (KAnJ, 85hy), the effect of different loss functions (KAnJ), additional quantitative evaluation on ImageNet (KAnJ), limitations in preserving semantic fidelity (85hy), and the use of a fixed text condition (1GUs).

After reviewing the rebuttal, most of these concerns were addressed through additional clarifications and experiments. Following the rebuttal, all reviewers would likely converge to positive ratings (6, 6, 6, 6). The approach is simple and effective, and the ability to leverage pretrained diffusion priors without retraining while enabling efficient single-step inference makes the method practically appealing. Given the overall contribution and the positive reviewer feedback, the AC leans toward recommending Accept.

**Reviewer Concerns:**

In response to the reviewers’ comments, the authors provided additional experiments, discussions, and results in the rebuttal and the revised paper. In particular, they clarified the motivation and positioning of the method relative to prior work, discussed the effectiveness of LPIPS-based supervision for timestep prediction, examined the impact of different loss functions, provided additional quantitative results on ImageNet, and discussed limitations related to semantic fidelity and fixed text conditioning.

Overall, most of the concerns listed above were sufficiently addressed, and no major outstanding issues remain at this stage.

**Reviewer Scores:**

Reviewers 85hy, 4wwB, and 1GUs gave positive initial ratings of 6 and did not raise new major concerns afterward, thus their scores would be expected to remain the same. Reviewer KAnJ (initial score: 2) noted during the discussion that he would raise his score. Overall, all reviewers appear to converge at positive ratings of 6.

---

### Decision · Program_Chairs · 2026-01-26

Accept (Poster)